# Modeling alpha-synuclein pathology in a human brain-chip to assess blood-brain barrier disruption

Iosif Pediaditakis [1,9,12✉], Konstantia R. Kodella [1,11], Dimitris V. Manatakis [1,11], Christopher Y. Le[1], Chris D. Hinojosa[1], William Tien-Street[1], Elias S. Manolakos[2,3], Kostas Vekrellis [4], Geraldine A. Hamilton[1], Lorna Ewart [1], Lee L. Rubin [5,6,7] & Katia Karalis[1,8,10,12✉]

Parkinson's disease and related synucleinopathies are characterized by the abnormal accumulation of alpha-synuclein aggregates, loss of dopaminergic neurons, and gliosis of the substantia nigra. Although clinical evidence and in vitro studies indicate disruption of the Blood-Brain Barrier in Parkinson's disease, the mechanisms mediating the endothelial dysfunction is not well understood. Here we leveraged the Organs-on-Chips technology to develop a human Brain-Chip representative of the substantia nigra area of the brain containing dopaminergic neurons, astrocytes, microglia, pericytes, and microvascular brain endothelial cells, cultured under fluid flow. Our αSyn fibril-induced model was capable of reproducing several key aspects of Parkinson's disease, including accumulation of phosphorylated αSyn (pSer129-αSyn), mitochondrial impairment, neuroinflammation, and compromised barrier function. This model may enable research into the dynamics of cell-cell interactions in human synucleinopathies and serve as a testing platform for target identification and validation of novel therapeutics.

[1] Emulate Inc., 27 Drydock Avenue, Boston, MA, USA. [2] Department of Informatics and Telecommunications, National and Kapodistrian University of Athens, Athens, Greece. [3] Northeastern University, Bouvé College of Health Sciences, Boston, MA, USA. [4] Biomedical Research Foundation of Academy of Athens, Athens, Greece. [5] Department of Stem Cell and Regenerative Biology, Harvard University, Cambridge, MA, USA. [6] Harvard Stem Cell Institute, Cambridge, MA, USA. [7] Broad Institute of Massachusetts Institute of Technology and Harvard, Cambridge, MA, USA. [8] Endocrine Division, Children's Hospital, Harvard Medical School, Boston, MA, USA. [9] Present address: Serqet Therapeutics, Inc. 55 Cambridge Parkway, Suite 800E, Boston, MA 02142, USA. [10] Present address: Regeneron Pharmaceuticals, 777 Old Saw Mill River Rd, Tarrytown, NY 10591, USA. [11] These authors contributed equally: Konstantia R. Kodella, Dimitris V. Manatakis. [12] These authors jointly supervised this work: Iosif Pediaditakis, Katia Karalis. ✉email: spediaditakis@serqettx.com; katia.karalis@regeneron.com

In Parkinson's disease (PD) and related synucleinopathies, the accumulation of alpha-synuclein (αSyn) plays a key role in disease pathogenesis. Assessment of the pathology in post-mortem brains from PD patients has demonstrated abnormal inclusions, enriched in misfolded and aggregated forms of αSyn, including fibrils[1,2]. These findings together with a wealth of experimental data, support the hypothesis for a central role of αSyn aggregation in the formation of the Lewy bodies and, therefore in the pathogenesis of synucleinopathies[3–5]. Recently, αSyn has been identified in body fluids, such as blood and cerebrospinal fluid[6,7], and has been postulated that it is also produced by peripheral tissues[8,9]. However, the ability of αSyn to cross the blood-brain barrier (BBB) in either direction and its potential contribution to the endothelial dysfunction described in patients with PD[10–13], remains unclear.

Experimental models of PD, such as animal models[14,15] or conventional cell culture systems[16–18], have advanced our understanding of the role of αSyn and its aggregated forms in the development of the disease and the induction of neuronal toxicity. However, these models have not been able to uncover the dynamics of the specific interactions between the brain parenchymal cells and the BBB, in normal or pathological states. Animal models of synucleopathies have so far shown a minimal ability for translation of their findings to human patients, including the cascade of tissue responses, that require specialized imaging and frequent sampling[19]. Conventional cell culture systems, including co-culture in Transwells, also have limitations such as difficulty to maintain nutrient concentrations, lack of fluid flow, and compromised ability, if any, to recapitulate the cell–cell interactions and cytoarchitecture at the neurovascular unit[20,21].

Recently, microengineered Organs-on-Chips[22,23] have been successfully developed for multiple complex organs, including intestine, lung, liver, heart, and brain[24–30]. Organs-on-Chips enable the recreation of a more physiological mircroenvironement, including co-culture of relevant cells on tissue-specific extracellular matrices (ECM), exposure to continuous flow, and in vivo-relevant mechanical forces such as the fluidic sheer stress. Lately, there have been several approaches to model the BBB in a chip, with models designed to reconstitute the cerebrovascular interface, however, they often do not include combinations of all key cell types, such as region-specific neurons, astrocytes, and microglia, critical to reconstruct the complex physiology of the neurovascular unit[31–35].

In the present study, we describe an approach where we have populated the Brain-Chip with human dopaminergic neurons a characteristic cell type of the substantia nigra, the area predominantly affected in PD (referred to here as the "Substantia Nigra Brain-Chip"). Our Substantia Nigra Brain-Chip recreates the vascular-neuronal interface, and it is populated with human iPSC-derived brain endothelial cells, pericytes, astrocytes, microglia, and dopaminergic neurons. In order to model states of exposure to abnormal αSyn aggregation and confirm the capability of the Substantia Nigra Brain-Chip to generate clinically relevant endpoints, we reconstructed a model of synucleinopathy by introducing human αSyn pre-formed fibrils (PFFs or "αSyn fibrils") within the brain channel. We provide evidence that this model replicates pathological hallmarks observed in human PD brains, including pSer129-αSyn accumulation, mitochondrial dysfunction, and progressive neuronal death[36]. In parallel, we show activation of astrocytes and microglia, in line with the active inflammatory process in the substantia nigra in patients with PD[37]. Further, we provide evidence that the worsening in brain pathology over time impacts the whole neurovascular unit, as evidenced by the compromised BBB permeability.

When taken together, these data suggest that the human αSyn fibril-induced disease model we have established in the Substantia Nigra Brain-Chip provides a model for recapitulating complex pathophysiological features of PD, including the BBB dysfunction. This model can be employed as a platform for target identification, target validation, and for evaluating the efficacy of therapies against PD and other synucleinopathies.

## Results

**Reconstitution and characterization of a human Substantia Nigra Brain-Chip.** For the Substantia Nigra Brain-Chip, we leveraged our previously described Organs-Chip design[25], which consists of two microfluidic channels fabricated from polydimethylsiloxane (PDMS) elastomer separated by a thin (50 μm) PDMS membrane containing multiple pores (7 μm diameter, 40 μm spacing). Each channel has dedicated inlet and outlet ports for the seeding of the human cells, that are maintained under controlled, independent for each channel, laminar flow applied (Fig. 1a). The membrane that separates the two channels is coated on both sides with a tissue-specific ECM cocktail, optimized for the Brain-Chip to contain collagen type IV, fibronectin, and laminin. First (D0), we seeded in the brain channel human iPSC-derived dopaminergic (DA) neurons derived from a healthy donor, as well as human primary brain astrocytes, microglia, and pericytes at respective seeding densities, as described in the Materials section. The next day (D1), we seeded human iPSC-derived brain microvascular endothelial cells (HBMECs) on the opposite surface of the membrane (Fig. 1b). Glia (astrocytes and microglia) and pericytes cultured in the brain channel support the proper development and maintenance of the BBB function, as previously reported[34,38,39]. To the best of our knowledge (based on data provided by the selling company) these donors do not carry any of the mutations in genes and SNPs associated with PD susceptibility. Further analysis of the transcriptomic profiling and SNPs genotyping of the Brain-Chip, showed expression of PD-related genes[40] at levels similar to those in the healthy adult human substantia nigra tissue. Common LRRK2 polymorphisms, such as rs34637584 and rs33949390, were not detected in any of the donors used (Supplementary Fig. 1a, b).

The vascular and brain channels were perfused with endothelial cell medium supplemented with 2% platelet-poor plasma-derived serum, and specific Dopaminergic Neurons Media, respectively (see "Methods"). We used cell morphology and metabolic activity assays (see "Methods" for details), to confirm that the culture conditions and the DA neurons medium used, were able to support the cellular composition of the brain channel throughout the length of the study (Supplementary Fig. 2a–d). The Substantia Nigra Brain-Chip was maintained for 2 days (D1-D2) in static culture, to promote the formation of the endothelial lumen and acclimate the cells to the microenvironment, before switched to continuous medium flow at 60 μL/h. Double-label immunofluorescence with antibodies against tyrosine hydroxylase (TH) and microtubule-associated protein 2 (MAP2) after 8 days in culture, revealed the vast majority of neurons as TH-positive (~80%), confirming their midbrain dopaminergic neuronal identity (Fig. 1c). Furthermore, the majority of the TH-positive neurons were positive for both FOXA2 (day 2: 89.5 ± 1.7 and day 8: 92.8 ± 1.6%), and LMX1A (day 2: 88.3 ± 0.8 and day 8: 90 ± 1.5%) over the course of the culture, indicative of their midbrain floorplate origin (Supplementary Fig. 2e, f). Similarly, the other cells of the co-culture in the brain channel of the Substantia Nigra Brain-Chip were stained positive on D8 of the culture for the cell-specific markers glial fibrillary acidic protein (GFAP; astrocytes), or transmembrane protein 119 (TMEM119; resting microglia), or

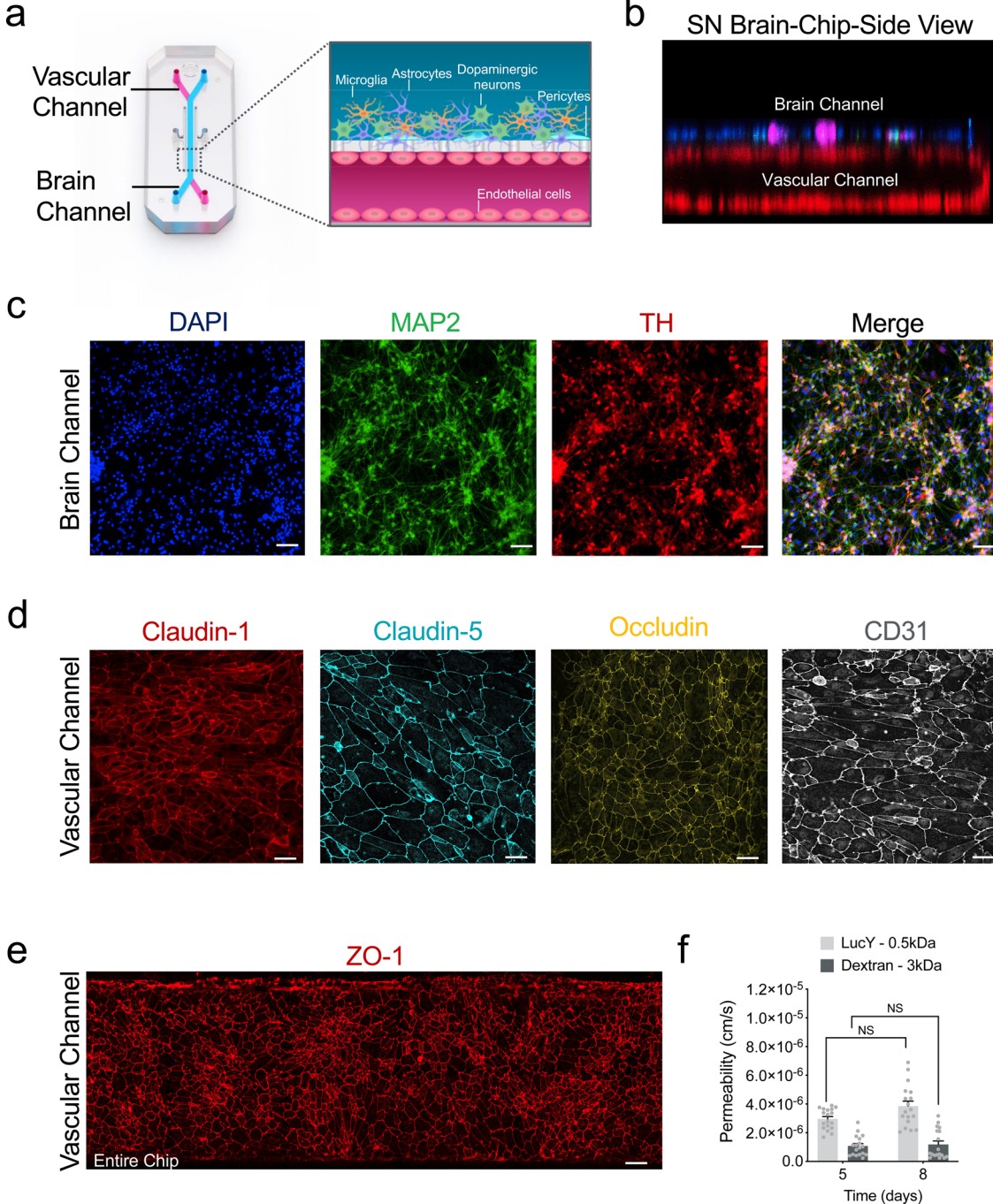

**Fig. 1 Reconstitution of the human substantia nigra Substantia Nigra Brain-Chip. a** Schematic depiction of the Substantia Nigra Brain-Chip of a two-channel microengineered chip with iPSC-derived brain endothelial cells cultured on all surfaces of the lower vascular channel, and iPSC-derived dopaminergic neurons, primary human brain astrocytes, microglia, and pericytes on the upper surface of the central horizontal membrane in the upper brain channel. **b** 3D reconstruction of a confocal z-stack showing the organization of all five cell types in the Substantia Nigra (SN) Brain-Chip. **c** Representative image of iPSC-derived dopaminergic neurons that are stained with DAPI (blue), microtubule-associated protein 2 (green, MAP2), tyrosine hydroxylase (red, TH), and a merged image on day 8. Scale bars: 100 μm. **d** Immunofluorescence micrographs of the human brain endothelium cultured on the vascular channel of Brain-Chip for 7 days post-seeding (D8) labeled with Claudin-1 (red), Claudin-5 (cyan), Occludin (yellow), and CD31 (white). Scale bars: 100 μm. **e** Immunofluorescence micrographs demonstrate high levels of expression of zonula occludens-1 (red, ZO-1) across the entire endothelial monolayer. Scale bars: 100 μm. **f** Quantitative barrier function analysis via apparent permeability to 3 kDa fluorescent dextran, and 0.5 kDa lucifer yellow (LucY) crossing through the vascular to the neuronal channel on day 5 and 8. Statistical analysis is two-way ANOVA with Tukey's multiple comparisons test ($n = 18$ independent chips, NS not significant). Error bars represent mean ± SEM.

proteoglycans (NG2; pericytes (Supplementary Fig. 2c). Next, we assessed the secreted dopamine levels via enzyme-linked immunosorbent assay (ELISA) to confirm the functionality of the dopaminergic neurons in the Brain-Chip. As shown (Supplementary Fig. 2g), the secretion of dopamine was stable over the length of the culture. Notably, the dopamine levels were significantly higher in the Brain Chip as compared to the monoculture of iPSC-derived dopaminergic neurons, indicative

of the effect of a physiologically relevant cellular environment, such as of the Brain-Chip, in the functional maturation of the DA neurons.

Development of tight junctions in the endothelial monolayer in the vascular channel of the Substantia Nigra Brain-Chip was confirmed based on the expression of Claudin-1, Claudin-5, Occludin, ZO-1 as well as the cell–cell adhesion protein CD31 (Fig. 1d, e), as previously described for the cerebral endothelial cells of the human BBB[34,41]. The Substantia Nigra Brain-Chip sustained its barrier integrity for up to 7 days in culture under continuous perfusion (D8), as assessed by low passive diffusion of dextran Cascade Blue (Mw: 3 kDa), and lucifer yellow (Mw: 0.5 kDa). Specifically, the apparent permeability of the BBB in the Substantia Nigra Brain-Chip was maintained at values within a range of $1–3 \times 10^{-6}$ cm s$^{-1}$ and $4–6 \times 10^{-6}$ cm s$^{-1}$, for dextran (3 kDa) and luciferin yellow (0.5 kDa) respectively, indicating the size-dependent transport across the BBB of the Substantia Nigra Brain-Chip (Fig. 1f). Notably, the low permeability of the Brain-Chip to dextran was comparable to previously reported in vivo values[42,43].

**Transcriptomic profiling of the Substantia Nigra Brain-Chip.** Next, we compared the global RNA-sequencing (RNA-seq) data of neurovascular units constructed in conventional cell cultures ($n = 4$), of Substantia Nigra Brain-Chips cultured under constant flow ($n = 4$), and of human adult brain-derived substantia nigra ($n = 8$) retrieved from the Genotype-Tissue Expression (GTEx) Portal[44]. The conventional cell cultures and the Substantia Nigra Brain-Chips were seeded using the same cell-type composition and subjected to the same experimental conditions (see "Methods"). We first performed differential gene expression (DGE) analysis between the Substantia Nigra Brain-Chip and the conventional cell cultures. To select the differentially expressed (DE) genes, we applied the following thresholds: adjusted (adj) $p$-value $< 0.05$ and |log2FoldChange|>1. Out of the 38,887 genes annotated in the genome, 1316 were significantly DE, with 646 and 670 genes up- and downregulated respectively, in the Substantia Nigra Brain-Chips (Fig. 2a). Then, we performed Gene Ontology analysis utilizing the Gene Ontology knowledgebase, to highlight the biological processes significantly enriched within these gene sets. Among the upregulated genes in the Substantia Nigra Brain-Chip samples, we identified functional gene sets that significantly clustered under 669 GO terms. These functional gene sets were part of brain-relevant biological processes, including synaptic transmission, ion transport, metabolic and immune processes, extracellular matrix organization, cell adhesion, tissue development, and stimuli-evoked responses (Fig. 2b). Compared to the Substantia Nigra Brain-Chip, the transcriptome of the conventional cell cultures was enriched in genes involved in cell division, microtubule cytoskeleton organization implicated in mitosis, and cell cycle processes (Fig. 2c). These findings indicate that in the Substantia Nigra Brain-Chip, the cells achieve a more mature and/or differentiated state compared to the cells in the conventional cell cultures, which seems to favor the cell proliferating state. These results are in line with previous studies showing that stem cell-based tissue models exhibit higher resemblance to the biological properties of the mature tissue when developed in Organs-on-Chips, as compared to conventional cell cultures[45–47].

Next, we performed additional DGE analysis, to determine the specific gene sets that may confer the closer similarity between the Substantia Nigra Brain-Chip and the adult substantia nigra tissue, as compared to the conventional cell cultures. For this purpose, we analyzed the differences between the Substantia Nigra Brain-Chip or the adult substantia nigra tissue and the conventional cell cultures (Substantia Nigra Brain-Chip versus conventional cell cultures and adult substantia nigra versus conventional cell cultures). We identified 1316 and 680 DE genes, respectively,

from each of the above comparisons, with 209 genes at the intersection of the two (Fig. 2d). We reasoned that these 209 DE genes, which were common for the Substantia Nigra Brain-Chip and the human adult substantia would provide insights to biological processes in human substantia nigra tissue that were maintained in the Substantia Nigra Brain-Chip. To get further insights into the biological processes enriched in this gene set, we performed Gene Ontology enrichment analysis. This analysis revealed that the 209 overlapping genes were associated with 25 significant GO terms. Notably, the biological processes enriched in this gene set were associated with essential functions such as secretion, transport, as well as tissue and system development (Supplementary Fig. 1c).

We next performed a direct comparison of the RNA-sequencing profiles by comparing DGE analyses between the following groups: (i) Adult Substantia Nigra vs. Brain-Chip and (ii) Adult Substantia Nigra vs. Conventional Cell Culture system. The first analysis revealed 566 significantly DE genes in adult substantia nigra, 186 up- and 380 downregulated. The second analysis revealed 682 DE genes in adult substantia nigra 205 up- and 477 downregulated (Supplementary Fig. 1d). In line with these findings, principal components analysis (PCA) shows clear separation between the samples of the: (i) conventional culture systems, (ii) Brain-Chip technology, and (iii) adult SN (Supplementary Fig. 1e) which indicates the transcriptomic differences between these groups. Although the numbers of the DE genes (absolute number) between the conventional culture system or the Brain-Chip as compared to adult substantia nigra were similar (Supplementary Fig. 1d), the expression levels of a set of 13 cell signature-specific genes characteristic of the mature tissue in the adult substantia nigra were much closer to those in the Brain-Chip rather than to those in the conventional cell culture systems (Fig. 2e). For each of these 13 genes, we compared the fold changes in expression levels (smaller |Log2(Fold Change)| values) in the following groups: (i) Adult Substantia Nigra vs. Brain-Chip and (ii) Adult Substantia Nigra vs. Conventional Cell Culture system (Fig. 2e).

**Establishment of an αSyn fibril model in the Substantia Nigra Brain-Chip.** Encouraged by the results of transcriptomic analysis detailed above, we then assessed whether the Substantia Nigra Brain-Chip would respond to abnormal, toxic protein species, like those found in synucleinopathies, as has been reported in experimental disease models or clinical disease. To this end, we used αSyn fibrils, a principal constituent of Lewy bodies shown to exert toxicity in DA neurons[48,49]. Analysis of αSyn monomers and αSyn fibrils revealed very low endotoxin levels as measured in the culture media using the LAL assay (Supplementary Fig. 3a). First, we characterized the capability of exogenously added fibrils to be accumulated and processed by the cells in the Substantia Nigra Brain-Chip. Human recombinant αSyn fibrils (4 µg/mL) were added in the culture medium of the brain channel under continuous flow, on Day 2 of the culture (Fig. 3a). Three- and 6-days upon-exposure to αSyn fibrils (D5 and D8 of the experiment, respectively), we assessed by immunostaining the abundance of phosphoSer129-alpha-synuclein (phospho-αSyn129), a post-translational modification characteristic for the pathogenic αSyn species[50,51]. Our results showed that exposure of the Substantia Nigra Brain-Chip to αSyn fibrils was sufficient to induce phosphorylation of αSyn in a time-dependent manner (Fig. 3b, c, and Supplementary Fig. 3b). Induction of phospho-αSyn129 was only evident following exposure to αSyn fibrils, as the same amount of αSyn monomer or PBS did not lead to the induction of detectable phospho-αSyn129 in the culture. Phosphorylation of αSyn in the brain channel was also confirmed by immunoblot

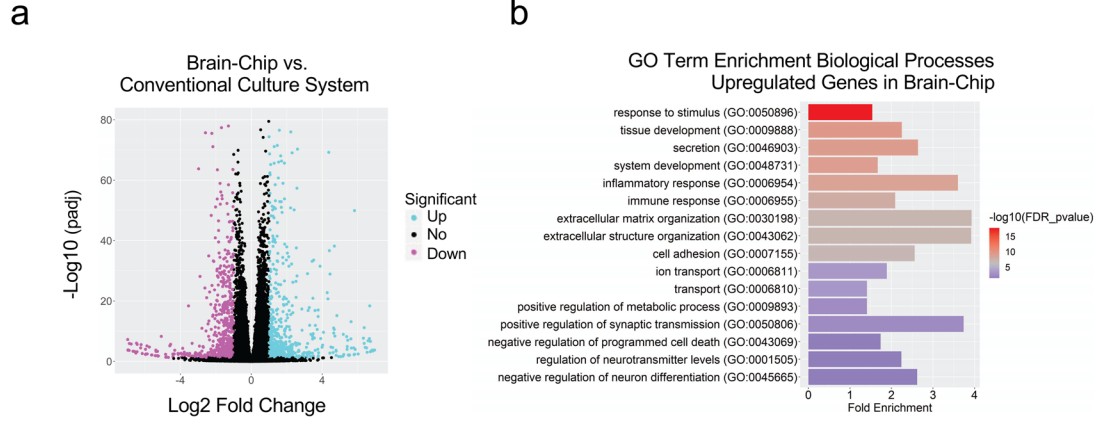

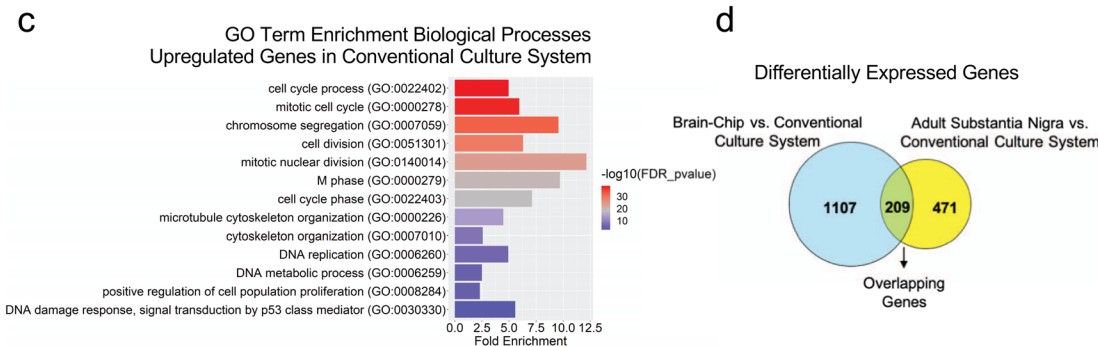

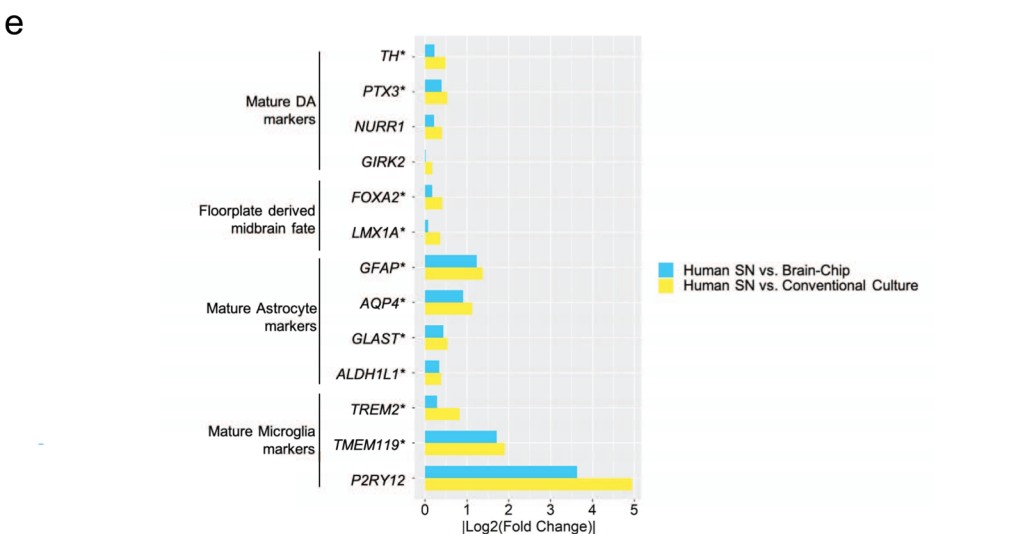

analysis at 6 days upon exposure to αSyn fibrils (Fig. 3d). Immunofluorescence staining showed phospho-αSyn129 accumulation at three- and 6-days post exposure to αSyn fibrils within TH positive neurons (Supplementary Fig. 3c, and Fig. 3e), astrocytes and microglia (Supplementary Fig. 3d, e and Fig. 3f, g), in agreement with the previous reports[52]. Taken together, these results demonstrate the specific effect of αSyn fibrils in the induction of phospho-αSyn129 pathology in our model.

**Effects of αSyn fibrils in mitochondria and ROS production in the Substantia Nigra Brain-Chip.** Emerging evidence indicates the critical role of mitochondrial dysfunction and the increase in

reactive oxygen species in the development of neurodegenerative diseases, including sporadic PD[53]. To assess the mitochondrial membrane potential in the cells in the Substantia Nigra Brain-Chip, we used JC-1, a staining probe for detection of mitochondrial damage. JC-1 in the form of a green monomer enters the cytoplasm and accumulates in the healthy mitochondria, where it forms numerous red J-aggregates. The transition of fluorescence signal from red to green indicates loss of mitochondrial membrane potential, as in cases of significant mitochondrial damage[54]. Exposure to αSyn fibrils led to lower intensity of red and increase of green fluorescence, in a time-dependent manner. Reduction in the red-to-green fluorescence intensity ratio was only found in the αSyn fibrils-exposed but not in the monomeric αSyn species-

**Fig. 2 Differentially Expressed (DE) genes and enriched gene ontology categories in Substantia Nigra Brain-Chip or conventional cell cultures as compared to the adult substantia nigra. a** The volcano plot resulted by the differentially gene expression (DGE) analysis between Substantia Nigra Brain-Chip and conventional cell cultures. For the selection of the DE genes we used the following thresholds: adjusted (adj) p-value < 0.05 and | Log2(foldchange)| >1. The identified up- (down-) regulated genes are highlighted in cyan (magenta) color respectively. Sample sizes were as follows: Substantia Nigra Brain-Chip, n = 4, conventional cell culture system, n=4. The p-values attained by the two-sided Wald test and corrected for multiple testing using the Benjamini and Hochberg method. **b, c** List of biological processes identified by Gene Ontology (GO) enrichment analysis using the up- and downregulated genes respectively resulted by the differentially gene expression analysis between Substantia Nigra Brain-Chip and conventional cell cultures. **d** DGE analysis identified up- and downregulated genes in Substantia Nigra Brain-Chip compared to conventional cell cultures (cyan circle), and human adult substantia nigra compared to conventional cell cultures (yellow circle). Sample sizes were as follows: Substantia Nigra Brain-Chip, n = 4, Conventional cell culture system, n = 4, and adult substantia nigra, n = 8 (independent biological specimens). Culture in Brain-Chips and conventional cell cultures were done in parallel. Samples were collected and processed for analyses 8 days post-seeding (D8). **e** Gene expression pattern analysis between the following groups: i) Adult Substantia Nigra vs. Brain-Chip and ii) Adult Substantia Nigra vs. Conventional Cell Culture system. In the Brain-Chip, most of these genes (those with an asterisk) exhibit statistically significantly smaller fold changes (P < 0.05) in their expression levels as compared to the conventional cell culture system. Statistical analysis is two-sided Wald test and corrected for multiple testing using the Benjamini and Hochberg method. The adjusted P value per gene is contained in Supplementary Table 1.

exposed Substantia Nigra Brain-Chips (Fig. 4a, b, and Supplementary Fig. 4a). Next, we measured the intracellular ROS levels in the brain channel of the Substantia Nigra Brain-Chip, on day 8 of the culture, using CellROX reagent. As shown (Fig. 4c, d), exposure to αSyn fibrils led to a significant increase in ROS production, as compared to exposure to αSyn monomers. Thus, we conclude that αSyn fibrils compromise mitochondrial function in the Substantia Nigra Brain-Chip, in a time dependent manner.

**αSyn fibrils induce cell death and neuroinflammation in the Substantia Nigra Brain-Chip.** Several studies have shown that αSyn fibrils initiate a series of secondary processes leading to neuroinflammation, neurodegeneration, and cell death[48,55]. We first questioned whether the cells in the Substantia Nigra Brain-Chip would respond to αSyn fibrils by induction of apoptosis. Three days (experimental D5) following exposure to αSyn, either monomeric or fibrillar, there was no effect in cell survival, as reflected by the similar percentage of live cells in all the experimental groups. In contrast, 6 days post exposure (experimental D8), there was significant reduction of the live cells in the Substantia Nigra Brain-Chip exposed to αSyn fibrils compare to αSyn monomers or PBS (50.63 ± 3.9 vs 87.02 ± 0.87 vs 91.2 ± 1.05) (Supplementary Fig. 4b, c). Confocal immunocytochemical analysis using antibodies against MAP2, tyrosine hydroxylase (TH), and cleaved caspase-3 (CC3), confirmed the increase in caspase 3-positive dopaminergic neurons at day 6 post exposure to αSyn fibrils, as compared to those exposed to monomeric αSyn (Fig. 5a, b). Dopaminergic neuron apoptosis was also evaluated and confirmed by TUNEL staining (Supplementary Fig. 4d, e).

Next, we assessed the extent of the inflammatory response upon exposure to αSyn fibrils in the Substantia Nigra Brain-Chip. We performed quantitative analysis on the gene expression of markers for reactive astrocyte and activated microglia, that are upregulated in pathological conditions. In particular, we found that 6 days post exposure of the Substantia Nigra Brain-Chip to αSyn fibrils, both reactive astrocyte markers (*GFAP*, *VIMENTIN*, and *LCN2*) and activated microglia markers (*CD68* and *EDA*) were upregulated compared to the monomeric αSyn-treated group (Supplementary Fig. 5a). We also observed increased GFAP staining in the αSyn fibrils-exposed chips suggestive of reactive astrogliosis (Fig. 5c and Supplementary Fig. 5b), a component of the brain inflammatory response[37]. In parallel, there was activation of the microglia, as indicated by the increase in CD68 immunoreactivity (Fig. 5d and Supplementary Fig. 5c). Interestingly, a large number of activated microglia formed clusters around the apoptotic neurons (Supplementary Fig. 5d), suggesting that the microglia response may be linked to the neuronal injury. Finally, we investigate the effect αSyn fibrils on

the proliferation of glial cells. Immunocytochemical staining with the proliferation marker Ki67 showed no significant increase in proliferation following exposure to αSyn fibrils as compared to monomeric αSyn (Supplementary Fig. 5e, f). In line, there were no differences in the number of cells between the control and treated groups (Supplementary Fig. 5g). Thus, we conclude that the astrocytic and microglial responses to αSyn fibrils in our model are due largely due to changes in the existing resting glial cells, rather than due to the generation of new glial cells from glial precursors. As expected, the secreted levels of interleukin-6 (IL-6), and tumor necrosis factor-alpha (TNF-α) in the effluent of the neuronal channel, were significantly increased following exposure to αSyn fibrils versus monomeric αSyn (Fig. 5e, f), providing evidence that αSyn fibrils-induced pathology is associated with the development of neuroinflammation.

**BBB disruption in αSyn-associated PD pathology.** As the evidence on extraneuronal manifestations of PD is increasing, attention has been drawn to the effects of the disease on the BBB function. Measurable levels of αSyn have been identified in the brain and in the systemic circulation, and the current hypothesis is that they are implicated in disease onset and/or progression. The origin of peripheral αSyn remains a subject of discussion, as well as the possibility that αSyn could cross the BBB in either direction[12]. Recent data argue that there is BBB dysfunction in PD as proposed for other neurodegenerative diseases, and that the role of BBB in the pathogenesis and progression of PD may be pivotal[56]. Thus, we ran BBB permeability assays on the Substantia Nigra Brain-Chip upon exposure to αSyn fibrils, or αSyn monomers or PBS. Our data indicate significantly increased permeability to 160 kDa Immunoglobulins (IgG), 3 kDa dextran, and 0.5 kDa lucifer in the brain channel of the Substantia Nigra Brain-Chip 6 days after exposure to αSyn fibrils (Fig. 6a, b, and Supplementary Fig. 6a).

We next assessed whether increased BBB permeability was associated to disruption of the tight junction protein ZO-1. Staining of the endothelial cells in the vascular channel of the Brain-Chip with anti-ZO1 at D8, demonstrated tight junctions damage following exposure to αSyn fibrils, but not to monomeric αSyn (Supplementary Fig. 6b, c). Next, we examined the implication of mechanisms attributable to BBB impairment, such as changes in the expression of TOM20, which prevents α-synuclein-induced mitochondrial dysfunction[57], and ICAM-1, a key molecule in immune-mediated and inflammatory processes in endothelial cells[58]. Immunofluorescence analysis showed attenuated expression of the TOM20 and increased expression of ICAM-1 at 6 days (experimental D8) post exposure to αSyn fibrils compared to monomeric group (Supplementary Fig. 6b, c),

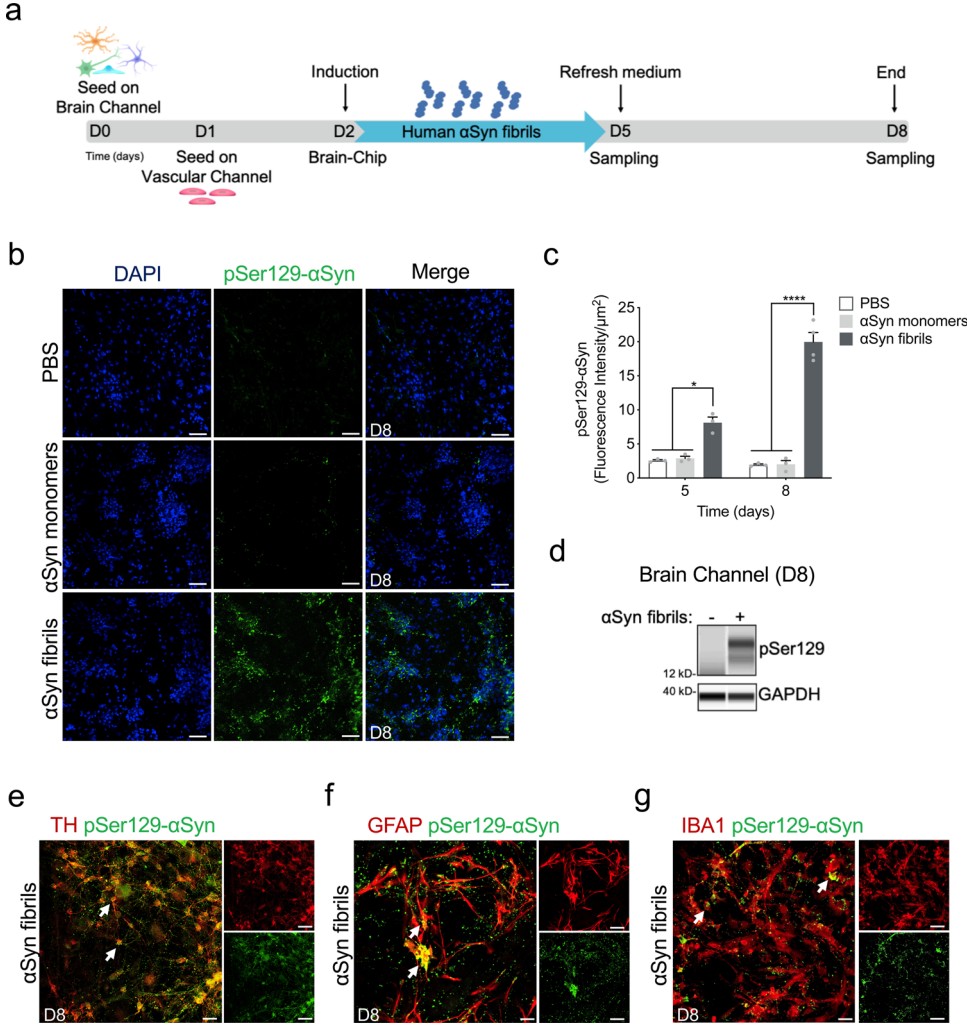

**Fig. 3 Pathological αSyn accumulation in the brain channel following exposure to human αSyn fibrils. a** Experimental design for assessing the effects of αSyn toxic aggregates (fibrils) in the Substantia Nigra Brain-Chip, including the seeding in the Brain-Chip, the timeline for medium changes, as well as sampling times. **b** Immunofluorescence micrographs show the accumulation of phosphorylated αSyn (green, phospho-αSyn129 staining; blue, DAPI) at day 6 post-exposure (D8). Pathology is absent in the brain channel following exposure to monomer or PBS. Scale bars: 100 μm. **c** Quantitative analysis of fluorescence intensity in each group at day 3 and 6 post-exposure (D5 and D8, respectively). Statistical analysis is two-way ANOVA with Tukey's multiple comparisons test (n = 3–4 independent chips with 3-5 randomly selected different areas per chip, *P = 0.0103, ****P < 0.0001 compared to monomeric group). Error bars represent mean ± SEM. **d** Western blotting analysis of cell lysates from the brain channel shows significant intracellular phosphorylation of αSyn at Ser129 (MW: 18 kDa) following exposure to αSyn fibrils, whereas there was no effect upon exposure to the PBS. For loading control, equal amounts of protein were immunoblotted with GAPDH antibody (MW: 37 kDa). **e** Confocal images of double immunostaining for phospho-αSyn129 (green) and tyrosine hydroxylase (red, TH), in the brain channel at day 6 post-exposure (D8) to αSyn fibrils (white arrow). Scale bars: 50 μm. **f, g** αSyn fibrils are taken up by astrocytes and microglia (white arrow), as evidenced by double immunostaining for phospho-αSyn129 (green) and either glial fibrillary acidic protein (red, GFAP) for astrocytes or ionized calcium-binding adaptor molecule 1 (red, IBA1) for microglia in the brain channel at day 6 post-exposure (D8) to αSyn fibrils. Scale bars: 50 μm.

suggesting a possible role for αSyn fibrils in BBB damage correlated with mitochondria dysfunction and inflammation in the vasculature. To check whether αSyn fibrils were directly deposited on the endothelial cells in the Brain-Chip, we performed immunofluorescent staining with phospho-αSyn129 in the vascular channel. Our results show that exposure of the Substantia Nigra Brain-Chip to αSyn fibrils-induced phosphorylation of αSyn localized in the endothelial cells in the vascular channel, 6 days post exposure to αSyn fibrils (Fig. 6c, d). The effective phosphorylation of αSyn in the vascular channel was confirmed by immunoblot analysis at 6 days post exposure to αSyn fibrils as compared to the PBS group (Supplementary Fig. 6d). These results indicate that αSyn fibrils affect the permeability of the BBB in the Brain-Chip, although our

experiment cannot distinguish between a direct role of αSyn fibrils or indirect via effects on other contributing molecules in causing BBB impairment.

To further characterize the endothelium in the Substantia Nigra Brain-Chip model and to determine whether the exposure to αSyn fibrils leads to transcriptomic changes in these cells, we ran RNA-seq analysis. PCA revealed differences in the transcriptome profiles between the two conditions, i.e., exposure to αSyn fibrils or to αSyn monomers (Fig. 6e). This analysis resulted in the identification of 1280 DE genes, either significantly upregulated (739 genes) or downregulated (541 genes) (Fig. 6f) in the αSyn fibril-exposed Substantia Nigra Brain-Chips. This set of 1280 DE genes includes several genes that have been implicated in BBB dysfunction in a number of diseases[59]. Multiple members

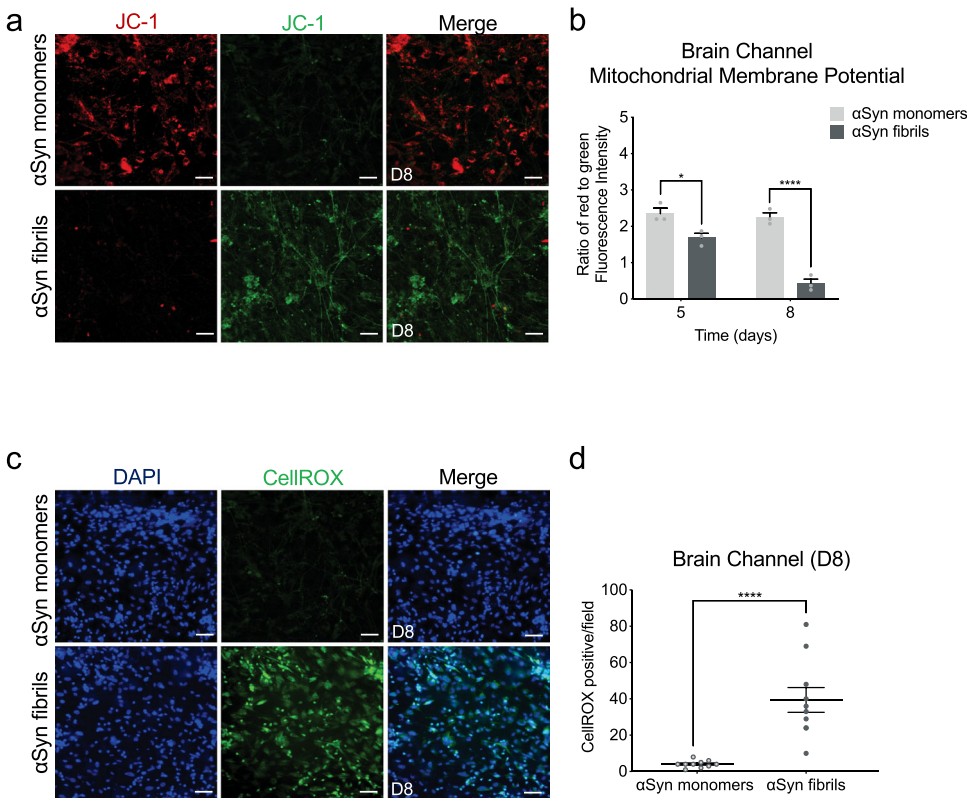

**Fig. 4 Reduction of mitochondrial activity and increase in ROS production in the αSyn fibril model. a** Mitochondrial membrane potential assessed by JC-1 staining in the brain side at day 6 post-exposure. Dual emission images (527 and 590 nm) represent the signals from monomeric (green) and J-aggregate (red) JC-1 fluorescence. Scale bars: 100 μm. **b** Quantitative analysis of the ratio of Red/Green fluorescence intensity in each group at day 3 and 6 post-exposure (D5 and D8, respectively). Statistical analysis is two-way ANOVA with Tukey's multiple comparisons test (n=3 independent chips with 3–4 randomly selected different areas per chip, *$P = 0.0389$, ****$P < 0.0001$ compared to monomeric group). Error bars represent mean ± SEM. **c** Representative images of reactive oxygen species (ROS) levels (green, CellROX) show higher levels of intercellular ROS in the cells of the brain channel exposed to αSyn fibrils than those exposed to αSyn monomer at day 8 post-exposure. Scale bars: 100 μm. **d** Quantification of the number of CellROX-positive events per field of view in each group. Statistical analysis is two-sided unpaired $t$-test ($n = 3$ independent chips with 3–4 randomly selected different areas per chip, ****$P < 0.0001$ compared to monomeric group). Error bars represent mean ± SEM.

of specific gene families were upregulated, such as extracellular proteases of the Serpin family (*SERPINA1*), collagens (*COL3A1*), centromere proteins (*CENPE*), and kinesins (*KIF15*). In addition, we identified multiple genes in the αSyn fibrils-exposed chips implicated in key cellular processes associated with PD pathogenesis (Table 1, Fig. 6g) such as autophagy, oxidative stress, mitochondrial function, inflammation, and vesicular trafficking, highlighting the potential for brain endothelial cells to contribute to molecular mechanisms and functional deficits in PD. Examples of associated genes include leucine-rich repeat kinase 2 (LRRK2)[60], synphilin-1 (*SNCAIP*)[61], monoamine oxidase A (*MAOA*)[62], complement 5 (*C5*)[63], and apolipoprotein A-1 (*APOA1*)[64]. The altered expression of PD-associated genes was confirmed by quantitative RT-PCR (qPCR) (Supplementary Fig. 6e). Other BBB-related genes with altered expression are low-density lipoprotein receptor-related protein 1B (*LRP1B*)[65] and ATP-binding cassette (ABC) transporters (*ABCB1*)[66]. The upregulation of the LRP1 gene is consistent with previous findings, where dysfunction of LRP1B has been associated with PD[67]. Further, positive and negative associations between specific ABCB1 haplotypes associated with P-glycoprotein activity PD incidence have been reported[66]. Endothelial genes downregulated upon exposure to αSyn fibrils included the tight junctions claudin gene family (*CLDN1*, *CLDN4*, and *CLDN9*), and the gap junction protein alpha 4 (*GJA4*). All the above suggest that the Substantia Nigra Brain-Chip could be used to assess novel treatments that

protect from vascular dysfunction or improve vascular remodeling in the brain.

**Trehalose ameliorates αSyn induced pathology**. To this purpose, we tested whether the disrupted BBB in the αSyn fibril-exposed Substantia Nigra Brain-Chip, could be restored by therapeutic agents targeting mechanisms for clearing of the accumulated αSyn protein. Recent reports of trehalose, a disaccharide approved by FDA, have shown beneficial effects against the accumulation of neurotoxic, aggregated proteins, and neurodegeneration[68]. The study of Hoffmann et al.[69] provided evidence that trehalose prevents or halts the propagation of αSyn pathology by targeting lysosomes. Also, studies in aged mice suggest that oral supplementation of the autophagy-stimulating disaccharide trehalose, restored vascular autophagy and ameliorated age-related endothelial dysfunction[70]. On the basis of these results, we speculate that trehalose may disturb lysosome integrity and function, which could subsequently hinder the BBB disruption induced by αSyn fibrils. In this regard, we administered trehalose (10 mM) in the Substantia Nigra Brain-Chip, via the vascular channel on experimental day 5, 3 days after adding αSyn fibrils (Fig. 7a).

To evaluate whether trehalose crosses the BBB and attenuates αSyn accumulation within αSyn-exposed cells, we first measured trehalose penetration in the brain channel. Using the trehalose assay kit, we found that more than 60% of trehalose had cross the

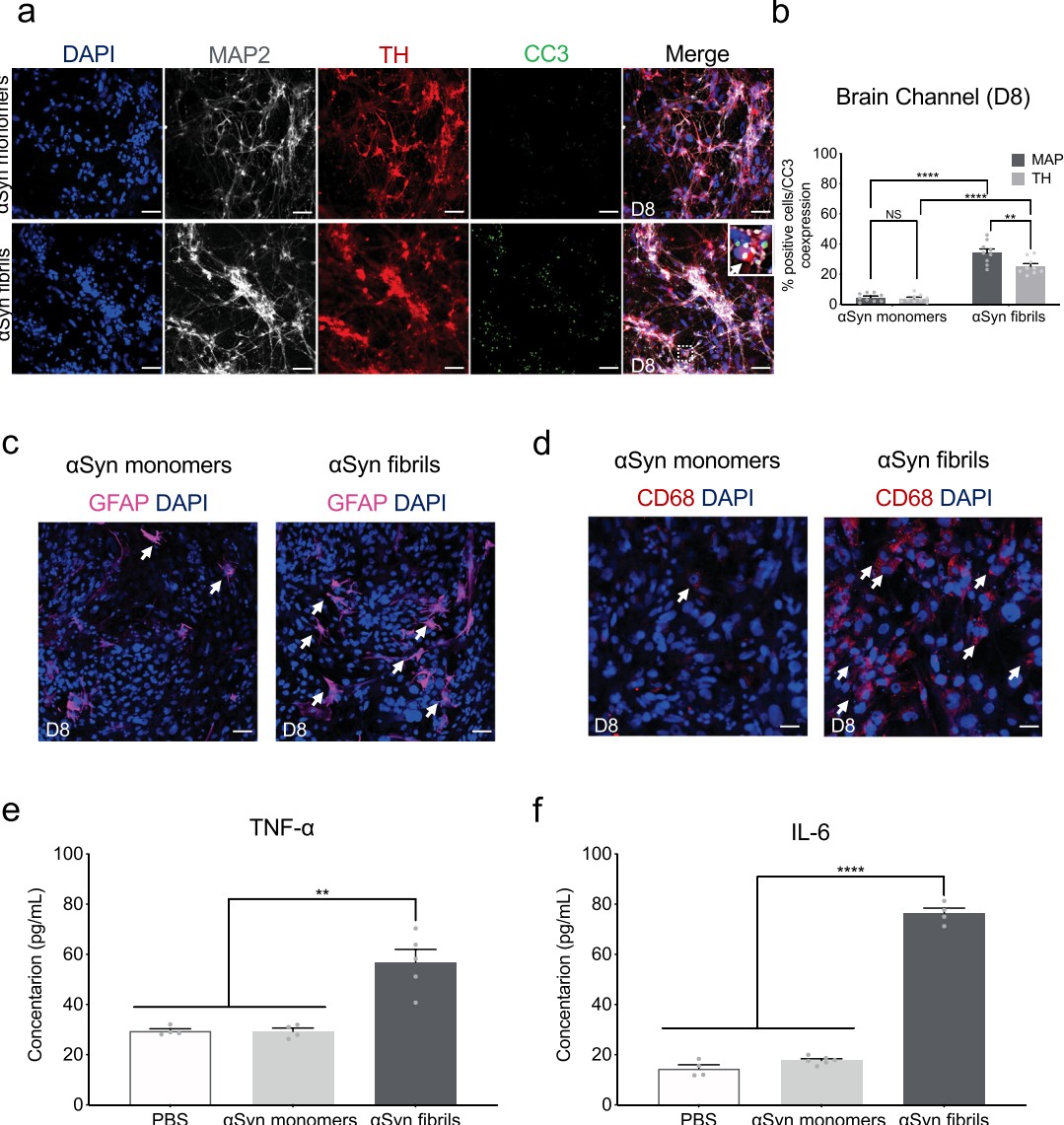

**Fig. 5 αSyn-induced caspase-3 activation and neuroinflammation. a** Representative merged images showing immunostaining for microtubule-associated protein 2 (gray, MAP2), tyrosine hydroxylase (red, TH), and Cleaved Caspase-3 (green, CC3) in the brain channel at 6-days post-exposure to αSyn fibrils or αSyn monomers. Scale bars: 100 μm. **b** Quantitation of the number of CC3 and MAP2- or TH-positive neurons. Statistical analysis by two-way ANOVA with Tukey's multiple comparisons test (3–4 randomly selected different areas per chip, $n = 3$ independent chips/experimental group, **$P = 0.0033$, ****$P < 0.0001$, NS not significant). Error bars represent mean ± SEM. Scale bars: 100 μm. **c** Immunostaining of the astrocyte marker glial fibrillary acidic protein (magenta, GFAP) demonstrated activation of astrocytes (white arrow) at day 6 post exposure to αSyn fibrils compared to monomeric αSyn. Scale bar, 100 μm. **d** Immunostaining of the microglial CD68 (red) demonstrated activation of microglia (white arrow) at day 6 post exposure to αSyn fibrils compared to monomeric αSyn. Scale bar, 100 μm. **e** The secreted levels of tumor necrosis factor-alpha (TNF-α) in the αSyn fibril model. Statistical analysis is two-sided unpaired $t$-test ($n = 6–7$ independent chips, **$P = 0.0023$). Error bars represent mean ± SEM. **f** The secreted levels of proinflammatory cytokine interleukin-6 (IL-6) in the αSyn fibril model. Statistical analysis is two-sided unpaired $t$-test ($n = 4–7$ independent chips, ****$P < 0.0001$). Error bars represent mean ± SEM.

BBB (Fig. 7b). As accumulation of phosphorylated αSyn in fibril-exposed cells was readily detectable in the brain channel (Fig. 3), we next validated the efficacy of trehalose in preventing lysosomal alteration and accumulation of aggregated αSyn. In agreement with previous findings[69], lysosomal cathepsin D activity decreased significantly in αSyn fibrils-exposed cells, indicative of impaired lysosomal activity (Supplementary Fig. 7a). Our results showed that trehalose prevented lysosomal alterations within αSyn-exposed cells in the brain channel (Supplementary Fig. 7a) and significantly decreased the abundance of phospho-αSyn129 compared to the non-treated chips (Fig. 7c and Supplementary Fig. 7b). Taken together, these findings

demonstrate the protective effects of trehalose on the αSyn-mediated impairment of lysosomes and on the clearance of accumulated αSyn in the cells within the brain channel. Next, we examined whether trehalose suppressed the secretion of proinflammatory cytokines detected in αSyn fibrils-exposed cells in the brain channel (Fig. 5e, f). We found that the TNF-α and IL-6 levels were significantly decreased in the brain channel's effluent following exposure to trehalose compared to the non-treated chips, in support of its protective effects against neuroinflammation (Fig. 7d).

At the vascular channel, trehalose exerted similar effects as it reduced the levels of phospho-αSyn129 (Fig. 7e and Supplementary

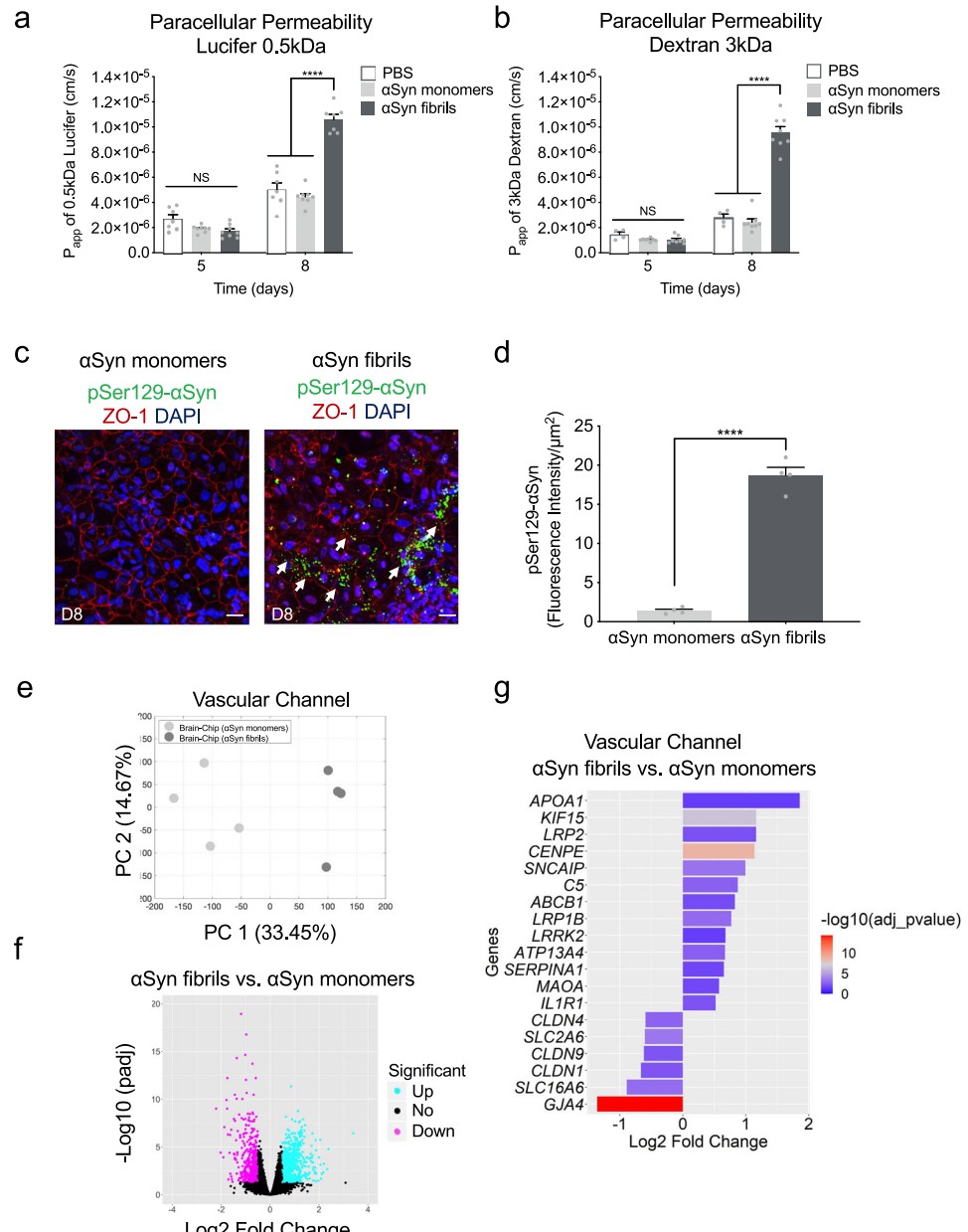

**Fig. 6 Blood-brain barrier dysfunction in the αSyn fibril model. a, b** Quantitative barrier function analysis via permeability to 0.5 kDa lucifer yellow and 3 kDa fluorescent dextran at day 5 and 8 following exposure to αSyn fibrils or αSyn monomers. Statistical analysis is two-way ANOVA with Tukey's multiple comparisons test ($n = 6$–9 independent chips, ****$P < 0.0001$ compared to monomeric group, NS not significant). Error bars represent mean ± SEM. **c** Immunofluorescence micrographs show accumulation of phosphorylated αSyn (green, phospho-αSyn129 staining; red, ZO-1; blue, DAPI) in the vascular channel of the Brain-Chip (white arrow), following treatment with αSyn fibrils, in contrast to lack of detected pathology in response to αSyn monomers. Scale bars: 100 μm. **d** Quantitative analysis of fluorescence intensity in each group at day 6 post exposure to αSyn fibrils or αSyn monomers. Statistical analysis is two-sided unpaired $t$-test ($n = 4$ independent chips with 3–5 randomly selected different areas per chip, ****$P < 0.0001$ compared to monomeric group). Error bars represent mean ± SEM. **e** Principal component analysis (PCA) generated using the RNA-seq data generated by the samples collected from the vascular channel of the Substantia Nigra Brain-Chip upon exposure to αSyn monomers or αSyn fibrils ($n = 4$ per condition). A 2D-principal component (PC) plot is shown with the first component along the $X$-axis and the second along the $Y$-axis. The proportion of explained variance is shown for each component. **f** The volcano plot shows the differentially expressed (DE) genes between αSyn fibrils and αSyn monomers, as identified by the RNA-sequencing analysis. For the selection of the DE genes we used the following thresholds: adjusted (adj) $p$-value < 0.05 and |Log2(foldchange)| > 0.5. The identified up- (down-) regulated genes are highlighted in cyan (magenta) color. Sample sizes were as follows: Brain-Chip (αSyn monomers), $n = 4$, Brain-Chip (αSyn fibrils), $n = 4$. The $p$-values attained by the two-sided Wald test and corrected for multiple testing using the Benjamini and Hochberg method. **g** A selection from the 739 upregulated and 541 downregulated genes identified after performing differentially gene expression (DGE) analysis between αSyn fibrils and αSyn monomers. The size of the bars indicates the log2(Fold-Change) of the corresponding gene expressions and the colors the statistical significance (False Discovery Rate; FDR adjusted (adj) $p$-value < 0.05) of the corresponding changes. Statistical analysis is two-sided Wald test and corrected for multiple testing using the Benjamini and Hochberg method.

**Table 1 Identification of multiple genes in the αSyn fibrils condition implicated in a variety of cellular processes.**

| Biological functions | Genes | Regulation |
|---|---|---|
| Mitochondrial oxidation | MAOA | Up |
| Inflammation | C5, IL1R1, SERPINA1 | Up |
| Autophagy and proteasome system | LRRK2, SNCAIP | Up |
| Vesicular trafficking | CENPE, KIF15 | Up |
| Endothelial active efflux | ABCB1 | Up |
| Lipoprotein receptors | LRP1B, LRP2, APOA1 | Up |
| Solute carrier-mediated transport | SLC16A6, SLC2A6 | Down |
| Tight junctions | CLDN1, CLDN4, and CLDN9 | Down |
| Gap junctions | GJA4 (commonly, Cx37) | Down |

Fig. 7c) and the secreted IL-6 and IFN-γ (Fig. 7f). IFN-γ levels were found significantly increased, following exposure to αSyn fibrils versus monomeric αSyn, only in the effluent of the vascular channel (Fig. 7d, f). Of note, IFN-γ levels are elevated in the blood plasma of PD patients[71] and reportedly increase BBB permeability[72,73]. These findings revealed a distinct, channel-specific profile of secreted proinflammatory cytokines in the αSyn fibrils-induced on chip-model of the disease, of potential significance for PD pathogenesis and targeting of the corresponding neuroinflammation.

After 72 h (experimental D8), we assessed BBB permeability by introducing 3 kDa dextran into the vascular channel (Fig. 7g). The trehalose-treated Substantia Nigra Brain-Chips showed significantly decreased BBB permeability (Fig. 7g) and rescuing of the damaged tight junctions following exposure to αSyn fibril (Fig. 7h). Notably, so far the effects of trehalose have only been evaluated in neuronal cell lines and animal models[69,74,75]. These results suggest that BBB-permeable molecules that prevent formation of protein aggregates and increase BBB integrity might have therapeutic potential for PD and that our model can be used to screen the efficacy of those molecules.

## Discussion

PD is characterized by an array of premotor and CNS symptoms, together with degenerative changes in the substantia nigra, which expand to more brain areas as the disease advances. Pathology findings reveal the existence of the characteristic Lewy bodies, which are proteinaceous aggregates containing αSyn[1–3]. Experimental models have significantly contributed to our comprehension of the pathogenesis of PD and other synucleinopathies, by demonstrating important aspects of αSyn biology, such as intracellular aggregation and neuronal death[76]. Despite the strong experimental and clinical evidence, the course of events driving the detrimental pathology in synucleinopathies remains unknown, as the diagnosis of the disease usually is made in later stages when the damage has already advanced. Furthermore, the existing animal models are limited in their relevance to human disease in terms of the mechanisms driving disease induction and progression. Given the complexity of the etiology and progress of synucleinopathies and the lack of in vivo models representative of the human disease, there is an unmet need for human cell-based models able to recreate the complex biology and elucidate cell–cell interactions driving the tissue pathology.

To address this need, we generated an engineered human Substantia Nigra Brain-Chip to capture the dynamic interactions in the human neurovascular unit, composed of iPSCs-derived brain endothelium and dopaminergic neurons, and primary astrocytes, pericytes, and microglia. A flexible, porus membrane, coated with tissue-relevant ECM, separates the endothelial from

the parenchymal cells cultured independently in specific medium, under continuous flow. This system is amenable to imaging and conventional endpoints used in both in vivo and in vitro studies. The system also enables frequent sampling of the effluent from either side of the membrane for assessment of barrier permeability and the characterization of the secretome at different time points.

Exposure of the Substantia Nigra Brain-Chip to αSyn fibrils led to progressive accumulation of phosphorylated αSyn and the associated induction of specific aspects of αSyn toxicity, such as mitochondrial dysfunction and oxidative stress. Compromised mitochondrial function, as reflected in mitochondrial complex I levels and development of oxidative stress, are central contributors in the neurodegenerative process in PD[53]. We also found that exposure of the Substantia Nigra Brain-Chip to αSyn fibrils, results in microglia activation, astrogliosis, and a time-dependent neuronal loss, as has been described in PD patients[37]. The dynamic, tunable microenvironment of the Substantia Nigra Brain-Chip, which includes physiological relevant mechanical forces and cytoarchitecture, including cell–cell interactions, may be the reason we were able to capture complex mechanisms driving key pathologies such as the gradual development of αSyn fibrils-induced toxicity. Another contributing factor might be that the cells on the Brain-Chip transition towards a more mature state, and recapitulate aspects of the brain responses that have not been captured by conventional cell culture systems. This hypothesis is supported by our transcriptomic data, and is in agreement with reports showing that maturation of neurons/astrocytes promotes the propensity of αSyn aggregation[49,77].

The first link between synucleinopathies and inflammation was provided by findings on activated microglia in the substantia nigra of PD patients[78]. Further, marked upregulation of *TNF-α* and *IL-6* mRNA levels were found in the substantia nigra of MPTP-treated animals compared to controls[79], as well as increased levels of inflammatory mediators in brain tissue from PD patients[80,81]. We similarly detected activation of the microglia and increased levels of secreted cytokines in the Substantia Nigra Brain-Chip effluent following exposure to αSyn fibrils. Although microglia is thought to be the key driver of the neuroinflammatory responses that propagate the neuronal cell death in PD, additional role(s) for microglia in the progress of synucleinopathies have been suggested[82]. We believe that the Substantia Nigra Brain-Chip provides opportunities to identify the exact interactions between microglia and other CNS cell types and how they could be targeted to modify the spread of αSyn pathology. A potential caveat of the current design is the lack of recruited peripheral immune cells, an important component of the disease. However, the perfusion capacity of this platform may be leveraged in the future to model the recruitment of disease-relevant immune cell subsets across the BBB, similar to the previous reports[83].

BBB dysfunction has been recently increasingly viewed as an inherent component of PD progression[10–13]. In PD animal models, including MPTP-treated mice[84] and 6-hydroxydopamine (6-OHDA)-treated rats[14], BBB disruption has also been found, in agreement with the clinical data. Additional studies have suggested that αSyn deposition increases BBB permeability[85] and PD development[86]. Despite the strong experimental and clinical evidence on the BBB disruption in PD, the underlying mechanisms remain unclear, whereas it is suggested that BBB involvement might even precede the dopaminergic neuronal loss in substantial nigra[87]. Finally, recent studies propose a peripheral origin to PD, suggesting the BBB's involvement in intestine-derived signaling that may induce brain pathology as an early mechanism in PD pathogenesis[88].

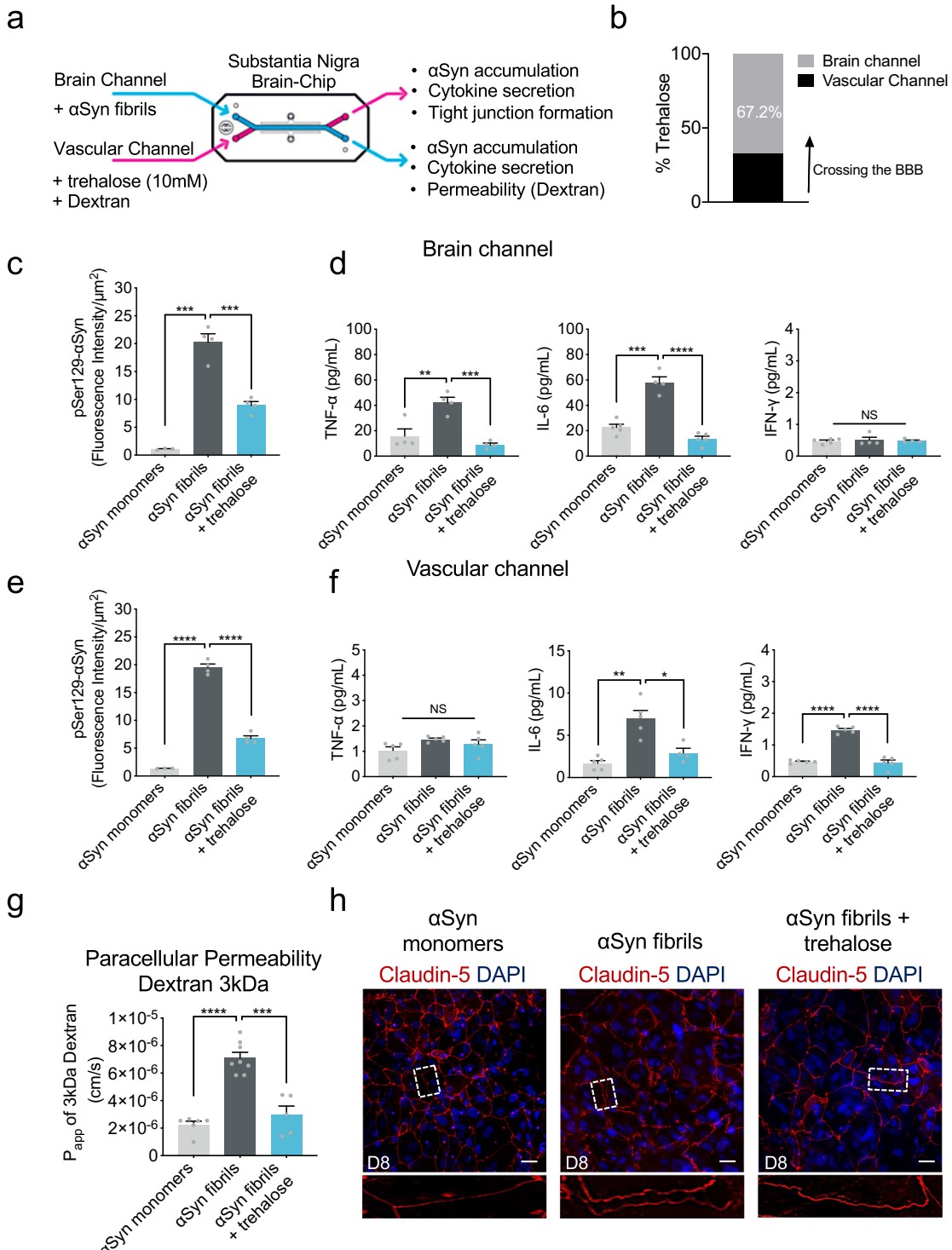

Our results show signs of tight junctions derangement and progressively compromised BBB permeability in response to αSyn fibrils. This is in line with previous studies showing deregulation of claudin as a key determinant of the BBB integrity and paracellular permeability[89]. Transcriptomic analysis of the BBB endothelium from the Substantia Nigra Brain-Chip revealed that αSyn fibrils alter the expression of genes associated with distinct biological processes implicated in PD, including autophagy, oxidative stress, mitochondrial function, and inflammation. Excitingly, control over the amount of αSyn accumulation by treatment with the autophagy inducer trehalose rescued the compromised BBB permeability and the derangement of the tight junctions, suggesting a prospective therapeutic approach for treating compromised BBB implicated in PD.

We recognize that the ideal Brain-Chip would be an isogenic model with all cell types originating from the same human individual, enabling most precise characterization of the cell–cell interactions and personalized medicine applications. However, an isogenic iPSC-based Brain-Chip requires robust and reproducible differentiation protocols for derivation of the several cell types of the CNS, validated and optimized by different groups. Even though the progress is tremendous, and some of the required

**Fig. 7 Effect of autophagic inducer, trehalose on the αSyn induced model. a** Schematic of the experimental design for the perfusion of trehalose and αSyn fibril in the Brain-Chip. **b** Percentage of trehalose crossing the blood-brain barrier (BBB) after 24 h of perfusion. **c** Quantitation of the accumulated phosphorylated αSyn at the brain channel based on the fluorescence intensity at day 8 in the αSyn fibril model with or without trehalose treatment. Statistical analysis by two-way ANOVA with Tukey's multiple comparisons test ($n = 3$–4 independent chips with 3–5 randomly selected different areas per chip, ***$P = 0.001$ compared to monomeric group, ***$P = 0.0009$ compared to αSyn fibrils). Error bars represent mean ± SEM. **d** Secreted levels of tumor necrosis factor-alpha (TNF-α), interleukin-6 (IL-6), and interferon gamma (IFN-γ) at the brain channel at D8. Statistical analysis by two-way ANOVA with Tukey's multiple comparisons test ($n = 4$ independent chips with 3–5 randomly selected different areas per chip, TNF-α: **$P = 0.0011$ compared to monomeric group, ***$P = 0.0003$ compared to αSyn fibrils, IL-6: ***$P = 0.0002$ compared to monomeric group, ****$P < 0.0001$ compared to αSyn fibrils, NS not significant). Error bars represent mean ± SEM. **e** Quantitation of the accumulated phosphorylated αSyn at the vascular channel based on the fluorescence intensity at day 8 in the αSyn fibril model with or without trehalose treatment. Statistical analysis by two-way ANOVA with Tukey's multiple comparisons test ($n = 4$ independent chips with 3–5 randomly selected different areas per chip, ****$P < 0.0001$). Error bars represent mean ± SEM. **f** Secreted levels of TNF-α, IL-6, and IFN-γ at the vascular channel at day 8 in the αSyn fibril model with or without trehalose treatment. Statistical analysis is two-way ANOVA with Tukey's multiple comparisons test ($n = 3$–4 independent chips with 3–5 randomly selected different areas per chip, IL-6: **$P = 0.0025$ compared to monomeric group, *$P = 0.0158$ compared to αSyn fibrils, IFN-γ: ****$P < 0.0001$, NS not significant). Error bars represent mean ± SEM. **g** Quantitative barrier function analysis via permeability to 3 kDa fluorescent dextran at day 8 in the αSyn fibril model with or without trehalose treatment. Statistical analysis is two-sided unpaired $t$-test ($n = 5$–8 independent chips, ****$P < 0.0001$ compared to monomeric group, ***$P = 0.0001$ compared to αSyn fibrils). Error bars represent mean ± SEM. **h** Morphological analysis of tight junctions in endothelial cells at day 8 in the αSyn fibril model with or without trehalose treatment. The junction protein expression of Claudin-5 was visualized by immunofluorescence staining with a Claudin-5 antibody and DAPI for cell nuclei. Dashed boxes show the endothelial junctions. Scale bars: 50 μm.

differentiation protocols already showed high robustness, the field has not yet reached the point where a reproducible, standardized isogenic model is available for disease modeling studies[90]. Variability between models remains a considerable issue, as exemplified in a study comparing the differentiation capacity of four different iPSC lines to isogenic BBB models, including endothelial cells and astrocytes[91]. A recent study successfully incorporated isogenic microglia-like cells with brain region-specific organoids from hiPSCs, however, these structures lacked the endothelial component required for BBB formation[92]. Therefore, for this study, we chose to employ well-characterized and established cell sources to develop stable and standardized assays that provide good cell quality and enable comparisons between experimental groups based on clinically relevant endpoints.

In conclusion, we report the generation of a Substantia Nigra Brain-Chip, that upon exposure to αSyn fibrils reproduces in vivo-relevant aspects of synucleinopathies. The Substantia Nigra Brain-Chip provides a promising model for the study of the specific disease mechanisms underlying this unmet medical need, including the dynamics of BBB dysfunction. Moreover, the chip may be useful to characterize the response to PD therapies and identify and evaluate associated biomarkers of the disease.

## Methods

**Cell culture**. Commercial human iPSC-derived dopaminergic neurons (iCell® Neurons) were purchased from Cellular Dynamics International (CDI; R1108) and maintained in complete maintenance media (iCell DopaNeurons Media). The cells have been characterized by CDI to represent a pure neuronal population with >80% pure midbrain dopaminergic neurons. Primary human astrocytes isolated from the cerebral cortex were obtained from ScienCell (Cat. No. 1800) and maintained in astrocyte medium (ScienCell). Primary human brain pericytes were also obtained from ScienCell (Cat. No. 1200) and maintained in the pericyte medium (ScienCell). Resting primary human brain microglia were purchased from ATCC (Cat. No. CRL-3304) and cultured according to the manufacturer's instructions. To confirm that the cell culture conditions applied were optimized, cell metabolic activity for each cell type in the brain channel medium was assessed using a (3-(4,5-dimethylthiazol-2-yl)-5-(3-carboxymethoxyphenyl)-2-(4-sulfophenyl)-2H-tetrazolium) MTS assay ($n = 6$), that measures the formazan product as an index of cell metabolic activity. The primary cells were used at passage 2–4.

**Brain microvascular endothelial cell differentiation of hiPSCs**. Human-induced pluripotent stem cells (hiPSCs) obtained from the Rutgers University Cell and DNA Repository (RUCDR; ND50028) were maintained on Matrigel-coated tissue-culture treated six-well culture plates (Corning) in mTeSR™1 (Stem Cell Technologies). The established hiPSC colonies displayed a normal karyotype in culture. For each independent experiment, we used the same cell passage (P49). Prior to differentiation, hiPSCs were singularized using Accutase® (Invitrogen) and plated

onto Matrigel®-coated six-well culture plates in mTeSR™1 supplemented with 10 μM Rho-associated protein kinase inhibitor Y27632 (ROCK inhibitor; Stem Cell Technologies) at a density between 25,000 and 50,000 cells cm$^{-2}$. Directed differentiation of hiPSCs was adapted from a previously reported protocol[39]. Briefly, singularized hiPSCs were expanded for 3 days in mTeSR™1, then were treated with 6 μM CHIR99021 (Stem Cell Technologies) in DeSR1. The DeSR1 medium is composed of DMEM/Ham's F12 (Thermo Fisher Scientific), 1× MEM-NEAA (Thermo Fisher Scientific), 0.5× GlutaMAX (Thermo Fisher Scientific), and 0.1 mM b-mercaptoethanol (Sigma). After 24 h, the medium was replaced by DeSR2 medium that is composed by DeSR1 plus 1× B27 (Thermo Fisher Scientific); the medium was refreshed every day for a period of another 5 days. On day 6, the medium was switched to hECSR1: hESFM (Thermo Fisher Scientific) supplemented with 20 ng/mL bFGF (R&D Systems), 10 μM all-trans retinoic acid (Sigma), and 1 × B27. The medium was not changed for 48 h. On day 9 the medium was switched to hESCR2: hECSR1 lacking RA and bFGF. On day 10, cells were dissociated with TrypLE™ (Thermo Fisher Scientific) and replated onto a human placenta-derived collagen IV/human plasma-derived fibronectin/human placenta-derived laminin-coated flasks. After 20 mins, the flasks were rinsed using a medium composed of human serum-free endothelial medium supplemented with 2% platelet-poor plasma-derived serum and 10 μM Y27632, as a selection step to remove any undifferentiated cells. Human brain microvascular endothelial cells (HBMECs) were then left in the same medium overnight to allow cell attachment and growth before seeded into the Organ-Chips.

**Brain-Chip microfabrication and Zoë® culture module**. Organ-Chips (Chip-S1®, Emulate, Inc. Boston, MA, USA) were used to recreate the human Brain-Chip. Chip-S1® is made of a poly(dimethylsiloxane) (PDMS) flexible elastomer. It consists of two channels (1 × 1 mm and 1 × 0.2 mm, "Brain" and "Vascular" channel, respectively) separated by a porous flexible PDMS membrane[25]. Flow can be introduced to each channel independently to continuously provide essential nutrients to the cells, while effluent containing any secretion/waste components from cells is excreted/collected on the outlet of each channel separately. This allows for channel-specific and independent analysis and interpretation of results. The Zoë® culture module is the instrumentation designed to automate the maintenance of these chips in a controlled and robust manner (Emulate, Inc.).

**Human Brain-Chip and cell seeding**. Prior to cell seeding, chips were functionalized using Emulate's proprietary protocols and reagents. Briefly, ER-1 (Emulate reagent: 10461) and ER-2 (Emulate reagent: 10462) are mixed at a concentration of 1 mg/mL before being added to the top and bottom microfluidic channels of the chip. The platform is then irradiated with high-power UV light having peak wavelength of 365 nm and intensity of 100 μJ/cm$^2$ for 20 min using a UV oven (CL-1000 Ultraviolet Crosslinker AnalytiK-Jena: 95-0228-01). After surface functionalization, both channels of the human Brain-Chip were coated with collagen IV (400 μg/mL), fibronectin (100 μg/mL), and laminin (20 μg/mL) overnight-both channels of the chip and then with DopaNeurons medium before seeding cells. A mixture of dopaminergic neurons, astrocytes, microglia, and pericytes is seeded in the upper brain channel of the Brain-Chips at the following concentrations: 2 million cells/mL for dopaminergic neurons, 2 million cells/mL for astrocytes, 0.1 million cells/mL for microglia, and 0.1 million cells/mL for pericytes. After cell seeding, the upper channel of the Brain-Chip was maintained in DopaNeurons medium and incubated overnight at 37 °C (Day 0). The following day (Day 1), the lower vascular channel was rinsed with human serum-free endothelial medium

supplemented with 2% platelet-poor plasma-derived serum, 10 μM Y27632, and then BMECs were seeded at a concentration of 16-20 million cells/mL to ensure the very tight endothelial monolayer found in the human BBB, and the chips were flipped immediately to allow BMECs to adhere to the ECM-coated part of the membrane. After 2 h incubation, the chips were flipped back to let the rest of BMECs sit on the bottom and sides of the channel to form a capillary lumen. The vascular channel of the Brain-Chip was maintained overnight. On Day 2, the Brain-Chips were connected to the Zoë® culture module and perfused continuously through the brain and vascular channel at a flow rate of 30 and 60 μl h⁻¹ respectively, using each channels' respective media.

**Conventional cell cultures**. The conventional cell cultures and the Brain-Chips were seeded using the same ECM composition as well as cell composition and seeding density. At the first experimental day (D0) the dopaminergic neurons, astrocytes, microglia, and pericytes were seeded on the apical side, followed by the seeding of the endothelial cells (D1) on the basolateral side of the 0.47 cm² Transwell-Clear permeable inserts (0.4-μm pore size). For the apical compartment we used iCell DopaNeurons Media, while for the basolateral compartment we used HBMEC medium. The cells maintained under static conditions throughout the duration of the experiment (D8). The culture medium was replaced daily in both compartments.

**Addition of exogenous alpha-synuclein to brain channel**. Human recombinant αSyn monomers and pre-formed fibrils were purchased from Abcam (Monomers; ab218819, Fibrils; ab218819), and were diluted in DopaNeurons Media to a final concentration of 4 μg/mL. On the day of use, αSyn fibrils were sonicated, and their activity was verified by Thioflavin T assay. Endotoxin levels were evaluated by the Limulus amebocyte lysate assay (Endotoxin Testing Services, Lonza Europe), and the amount expressed was negligible. For treatment, freshly prepared monomers and fibrils were used. On Day 2, the upper channel of the Brain-Chip was exposed to monomeric or fibrillar αSyn. After 3 days of exposure (D5), the medium was changed, and the culture was maintained using DopaNeurons Media (αSyn free) for three more days (D8). Effluents, lysates, and staining were collected/fixed at day 3- and day 6 post-exposure (D5 and D8 respectively), and were analyzed by a microplate reader, ELISA kits, western blot, and immunofluorescence microscopy.

**Permeability assays**. Culture medium containing 100 μg mL⁻¹ of dextran (3 kDa) and 20 μg mL⁻¹ of lucifer (0.5 kDa) tracers were dosed through the vascular channels for 24 h, and concentration of the dextran and lucifer tracers in the outlet samples from both vascular and brain channels was determined by using BioTek (BioTek Instruments, Inc., Winooski, VT, USA). Then, the following Eq. (1) was used to calculate Papp:

$$P_{app} = \frac{Q_R * Q_D}{SA * (Q_R + Q_D)} * \ln\left[1 - \frac{C_{R,0} * (Q_R + Q_D)}{(Q_R * C_{R,O} + Q_D * C_{D,O})}\right] \quad (1)$$

Here, SA is the surface area of sections of the channels that overlap (0.17cm²), $Q_D$ and $Q_R$ are the fluid flow rates in the dosing and receiving channels respectively, in units of cm³/s, $C_{D,O}$, and $C_{R,O}$ are the recovered concentrations in the dosing and receiving channels respectively, in any consistent units. IgG permeability was also evaluated after dosing the vascular channel and measuring the IgG content on the brain channel. Detection and quantitation of serum immunoglobulin G (IgG1; Abcam) was performed using the ELISA kit (Abcam), after 24 h of perfusion.

**Immunofluorescence microscopy**. Chips were fixed with 4% paraformaldehyde in PBS for 10 min and then washed with PBS. Immunostaining was performed after permeabilization in PBS with 0.1% Saponin and blocking for 30 min in 10% donkey serum in PBS with 0.1% Saponin. Immunostaining was performed with specific primary antibodies (Supplementary Table 2): rabbit anti-TH (1:500;abcam, ab6211), chicken anti-TH (1:100;abcam ab76442), goat anti-GFAP (1:300;abcam, ab53554), mouse anti-TH (1:100;Sigma, T2928), Rabbit anti-FOXA2 (1:200;Cell Signaling, 8186), rabbit anti-LMX1A (1:100;Sigma, ZRB1373), mouse ant-CD68 (1:100;abcam, ab955), mouse anti-ICAM-1 (1:100;R&D Systems, NET30), rabbit MAP2 (1:100;abcam, ab32454), mouse anti-TOM20 (1:100;abcam, ab56783), rabbit anti-IBA1 (1:50;FCDI, 019-19741), mouse anti-phosphoSer129 (1:100;abcam, ab184674), rabbit anti-phosphoSer129 (1:100;Cell Signaling, 23706), rabbit anti-ki67 (1:10;abcam, ab197234), rabbit anti-CD68 (1:100;abcam, ab213363), rabbit anti-Cleaved Caspase-3 (1:100;abcam, ab32042), rabbit anti-NG2 (1:100;abcam, ab83178), rabbit anti-TMEM119 (1:100;abcam, ab185333), rabbit anti-pSer129-αSyn (1:100;abcam ab51253), mouse anti-MAP2 (1:200;Thermo Fisher Scientific, MA512823), rabbit anti-CD31 (1:25;Thermo Fisher Scientific, RB-10333-P1), mouse anti-Claudin-1 (1:25;Thermo Fisher Scientific, 37-4900), mouse anti-Claudin-5 (1:50;Thermo Fisher Scientific, 35-2500), mouse anti-Occludin (1:100;Thermo Fisher Scientific, 33-1500), rabbit anti-ZO-1 (1:200;Thermo Fisher Scientific, 40-2200), mouse anti-ZO-1 (1:200;Thermo Fisher Scientific, ZO1-1A12). Primary antibodies were diluted in 10% donkey serum in PBS with 0.1% Saponin and incubated overnight on the Brain-Chip at 4 °C. After three PBS washes, cells were incubated with secondary antibodies in 10% donkey serum in PBS with 0.1%

Saponin (Alexa Fluor-488 (1:300;abcam, ab150105), Alexa Fluor- 568 (1:300;abcam, ab175470), and Alexa Fluor-647 (1:300;abcam, ab150135) for 1 h at room temperature when the primary antibodies are not conjugated, and cells were counterstained with DAPI nuclear stain. Cells were then washed with PBS three times and were visualized. Images were acquired with either an Olympus fluorescence microscope (IX83), Zeiss confocal microscope (AxiovertZ1 LSM880) or Opera Phenix (Perkin Elmer).

**Western blotting**. RIPA cell lysis buffer (50 mM Tris, pH 8.0, 150 mM NaCl, 5 mM EDTA, 1% NP-40, 0.5% sodium deoxycholate, and 1% SDS) supplemented with protease and phosphatase inhibitors (Sigma) was used for the extraction of total protein from either brain or vascular channel. The Auto Western Testing Service was provided by RayBiotech, Inc. (Peachtree Corners, GA USA). 0.1 mg/ mL sample concentration was loaded into the automated capillary electrophoresis machine. Phosphorylated αSyn at Ser129 was detected with a pSer129-α-syn antibody (1:100;Cell Signaling, 23706). Gluceraldehyde-3-phosphate dehydrogenase (GAPDH) antibody provided by RayBiotech (service library antibody) was used as the loading control. The uncropped WB images are provided in the source data file.

**Mitochondrial membrane potentials assay**. JC-1 probe was employed to evaluate the mitochondrial depolarization in cells seeded at the brain channel. Briefly, cells were incubated with 2 μM of JC-1 dye at 37 °C for 20 min and rinsed twice with PBS, then replaced in fresh medium. Finally, images were taken in the green and red fluorescence channel by confocal laser scanning microscopy imaging. The images were obtained at 488 nm excitation and 530 nm emission for green (JC-1 monomers) and 543 nm excitation and 590 nm emission for red fluorescence (JC-1 aggregates). Four frames per chip at ×10 magnification were selected for each treatment, and fluorescence intensity was measured using Fiji/ImageJ.

**Intracellular ROS measurement**. Intracellular ROS production was measured using CellROX® Green Reagent (Thermo Fisher Scientific) according to the manufacturer's protocol. At day 8, CellROX® reagent was added to the brain channel at a final concentration of 5 μM, and cells were incubated for 60 min at 37 °C in the dark, followed by triple washing with prewarmed PBS. Then, cells were examined with a confocal laser scanning microscope at an excitation/emission wavelength of 485/520 nm. Four frames per chip at ×10 magnification were selected for each treatment, and particles were counted using Fiji/ImageJ.

**Viability and cell death assay**. The cell viability was assessed using the LIVE/ DEAD staining kit (Thermo Fisher Scientific). The neuronal channel of the Brain-Chips was incubated for 30 min in PBS containing 1 μM calcein AM and 2 μM ethidium homodimer-1 (EthD1). The channel was then washed with PBS and imaged under a motorized fluorescent microscope (Zeiss confocal microscope). Four frames per chip at ×10 magnification were selected for each treatment, and particles were counted using Fiji/ImageJ. Data were expressed as the average live cells/total number of cells (sum of calcein-AM positive and ethidium homodimer positive). In order to confirm the efficiency and reliability of this assay, we set up a positive control (DMSO treatment) and negative control (no treatment) to do the parallel experiment with the αSyn treatment. In cell death analysis, TUNEL-positive cell density was measured for each treatment, and particles were counted using Fiji/ImageJ.

**Cytokine, dopamine, and trehalose measurements**. The levels of TNF-α, IFN-γ, IL-6, dopamine, and trehalose were measured by commercial ELISA kits (cytokines; Abcam, dopamine; BioVision, trehalose; MyBioSource) according to the manufacturers' instructions. The assays were performed in duplicate in 96-well plates.

**Cathepsin D activity**. Cathepsin D activity was determined using the Cathepsin D activity assay Kit (Abcam). Each channel of the Brain-Chip was lysed in 200 μl chilled CD cell lysis buffer supplied in the kit. Cells and the reaction mix were prepared according to manufacturer's protocol. The fluorescence derived from cathepsin D-mediated substrate cleavage was measured using a BioTek plate reader at excitation/emission = 328/460 nm.

**TaqMan genotyping assay**. Genomic DNA was isolated from each cell type using PureLink genomic DNA purification kit (Thermo Fisher Scientific) following the manufacturer's recommended protocol. DNA concentrations were determined using the NanoDrop 2000 UV/VIS spectrophotometer (Thermo Fisher Scientific). Genotyping by TaqMan was performed following as per manufacturer's instructions using 10 ng of DNA mixed with the TaqMan Genotyping Master Mix (Thermo Fisher Scientific) and custom TaqMan SNP assays (Thermo Fisher Scientific), Supplementary Table 3.

**RNA isolation, reverse transcription, and qPCR**. Each channel of the Brain-Chip was lysed using TRI Reagent® (Sigma- Aldrich, T9424) and RNA was isolated using

the Direct-zol RNA Purification Kit (Zymo Research, R2060). Genomic DNA was removed using the TURBO DNA-freeTM Kit (ThermoFischer, AM1907) and reverse transcription to cDNA was performed with the SuperScript IV Synthesis System (ThermoFischer, 18091050). TaqMan Fast Advanced Master Mix (Applied Biosystems, 4444963) and TaqMan Gene Expression Assays were used for the Quantitative Real-Time PCR (qPCR). The qPCR was detected on a QuantStudio 3 PCR System (Fisher Scientific). The target genes were assessed using commercially available primers (*GFAP*: Hs00909233_m1, *VIMENTIN*: Hs00958111_m1, *LCN2*: Hs01008571_m1, *CD68*: Hs02836816_g1, *EDA*: Hs03025596_s1, *APOA1*: Hs00163641_m1, *SNCAIP*: Hs00917422_m1, *LRRK2*: Hs01115057_m1, *MAOA*: Hs00165140_m1). The results were quantified by the comparative Ct method. Ct values for samples were normalized to the expression of the housekeeping gene (*18s*: Hs99999901_s1; Applied Biosystems). All primers are listed in Supplementary Table 4.

**RNA isolation and sequencing**. RNA was extracted using TRIzol (TRI reagent, Sigma) according to the manufacturer's guidelines. The collected samples were submitted to GENEWIZ South Plainfield, NJ, for next-generation sequencing. After quality control and RNA-seq library preparation the samples were sequenced with Illumina HiSeq $2 \times 150$ system using sequencing depth ~50 M paired-end reads/sample.

**RNA-sequencing bioinformatics**. Using Trimmomatic v.0.36 we trimmed the sequence reads and filtered-out poor quality nucleotides and possible adapter sequences. The remained trimmed reads were mapped to the homo sapience reference genome GRCh38 using the STAR aligner v2.5.2b which generated the BAM files. Using the BAM files, we calculated for each sample the unique gene hit-counts by applying the feature Counts from the subread package v.1.5.2. Note that only unique reads that fell within the exon region were counted. Finally, the generated hit-counts were used to perform differentially gene expression (DGE) analysis using the "DESeq2" R package (v.1.30.1) by Bioconductor.

**GO term enrichment analysis**. The gene sets resulted from the DGE analyses were subjected to gene ontology (GO) enrichment analysis. The GO terms enrichment analysis was performed using the gene ontology knowledgebase (Gene Ontology Resource http://geneontology.org/).

**GTEx human adult substantia nigra samples selection procedure**. GTEx Portal provides 114 RNA-seq samples for human adult substantia nigra. Eight representative samples out of the 114 samples were selected and combined with our eight samples from our Brain-Chip and conventional cell culture samples to generate a balanced dataset. For the selection of the eight representative samples we used the following criteria: (1) The samples belonged to donors who were reasonably healthy and they had fast and unexpected deaths from natural causes; and (2) Have the smaller transcriptomic distances[93] from the average transcriptomic expression profile of the samples that satisfy criterion (1). Next, we used the "removeBatchEffect" function of the "limma" R package (v.3.46.0)[94] in order to remove shifts in the means between our samples (Brain-Chip and conventional cell cultures) and the 8 human substantia nigra samples that we retrieved from GTEx portal[95]. The dataset was used to perform DGE analyses between the different conditions. For the DGE analyses, we used the 'DESeq2' R package (v.1.30.1) by bioconductor[96].

**Statistics and reproducibility**. All experiments were performed with controls (monomers or PBS) side by- side and in random order and they were reproduced for at least two times to confirm data reliability. All the attempts at replications were successful. The experiments in Figs. 1c–e; 3e–g; 5c–d; 7h and Supplementary Fig. 2a, c, e; 3b–e; 4a; 5d; 6b–c; 7b–c were performed in a minimal of two independent biological experiments. Western blots in Fig. 3d and in Supplementary Fig. 6d were performed once, using the antibodies and techniques listed in the "Methods" section. Statistical analyses were done using GraphPad Prism 9. All numeric results are shown as mean ± standard error of the mean (SEM) and represent data from a minimum of two independent experiments of distinct sample measurements. Analysis of significance was performed by using ANOVA with Tukey's multiple comparisons test or unpaired *t*-test depending on the data sets. Significant differences are depicted as follows: *$P < 0.05$, **$P < 0.01$, ***$P < 0.001$, and ****$P < 0.0001$.

**Reporting summary**. Further information on research design is available in the Nature Research Reporting Summary linked to this article.

## Data availability

All data generated or analyzed during this study are included in this published article (and its supplementary information files), and all additional information is available upon reasonable request to the authors. The microscope images have been deposited in Mendeley in a publicly available dataset (Pediaditakis et al.[97]). In addition, all the uncropped WB images are provided in the source data file. RNA-sequencing data have

been deposited in the Gene Expression Omnibus (GEO) under the accession number GSE168870. Source data are provided with this paper.

## Code availability

All the code used for the analysis in this report is derived from previously published reports. It is also explained and cited in the appropriate materials and methods or supplementary experimental procedures sections.

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

## Acknowledgements

We would like to thank Dr. Athanasia Apostolou for providing technical assistance. This work was supported by The Michael J. Fox Foundation for Parkinson's Research (16561; to I.P. and K.K.) and by the National Institute of Health, National Center for Advancing Translational Sciences (UG3TR002188; to C.D.H. and K.K.). The content is solely the responsibility of the authors and does not necessarily represent the official views of the National Institutes of Health. We also thank Brett Clair for scientific illustrations.

## Author contributions

I.P. designed, performed the experiments, and analyzed the data. K.R.K. helped perform experiments and interpret data. D.V.M. processed and analyzed the transcriptomic data, and incorporate the associated data in the manuscript. C.Y.L. helped perform experiments. C.D.H. provided insightful input on the engineering aspects of the project. W.T.S. helped with image analysis. E.S.M. was involved in the bioinformatic analysis and provided comments in the paper. L.E., K.V., and G.A.H. provided insightful comments on the findings through the project and reviewed the manuscript. L.L.R. provided critical feedback and reviewed the manuscript. K.K. and I.P. initiated, supervised the study, and wrote the paper with input from all others.

## Competing interests

I.P., K.R.K., D.V.M., C.Y.L., C.D.H., W.T.S., L.E., G.A.H., and K.K. are current or former employees of Emulate, Inc and may hold equity interests in Emulate, Inc. All other authors declare no competing interests.
