## [Peer Review File · Nature Communications]

Modeling Alpha-Synuclein Pathology in a Human Brain-Chip to Assess Blood-Brain Barrier DisruptionREVIEWER COMMENTS

Reviewer #1 (Remarks to the Author):

Pediaditakis et al. established a co-culture system modelling the human substantia nigra neurovascular interface, using Brain-Chip technology. Transcriptomic profile comparison suggested that Brain-Chip cultures achieved a more mature state as compared to conventional cultures. Moreover, Brain-Chip and adult substantia nigra showed common differentially expressed genes, as compared to conventional cultures. Exogenous aSyn fibril administration was used as a paradigm for validating Brain-Chip suitability for modelling Parkinson's disease pathomechanisms, and addressing effects on endothelial integrity. In addition to the expected phenotypes of phosphorylated aSyn accumulation, increased overall oxidative stress and neuroinflammation, Brain-Chip blood-brain barrier integrity was compromised by fibrils, and this could be rescued by trehalose. The described co-culture system is potentially highly relevant for studying Parkinson's disease biology, and can have translational value. Nevertheless, particularly the aspects of Brain-Chip suitability for studying dopaminergic vulnerability and neuron-glia interactions was not sufficiently studied in depth and requires elucidation.

Major concerns

1. Based on Figure 2d, the authors claim that the Brain-Chip recapitulates the gene expression pattern of adult substantia nigra. However, this remains to be addressed via direct Brain-Chip vs. adult substantia nigra RNA-sequencing profile comparison, or more suitable bioinformatics analyses including the 3 groups (Brain-Chip, adult substantia nigra, conventional cultures). Moreover, the global transcriptomic analysis is an important dataset of the study, but in order for the reader to interpret, the conventional cell culture should be defined and described in detail in the manuscript and materials and methods.
2. It is important to clarify whether the Brain-Chip is an advantageous setup for studying dopaminergic neuron vulnerability, and neuron-glia interactions in the paradigm of exogenous aSyn fibril administration. Specifically, Figure 5a-b, shows an overall decrease in neuronal density and increased caspase-3 activation, but this should be quantified for TH+ and TH- neurons. Moreover, the relationship between neuronal loss and neuroinflammation within the Brain-Chip should be elucidated. Which cells of the brain side show uptake of aSyn fibrils at earlier vs. later time points? With regards to glial responses (Figure 5c/d and Supplementary Figure 3c/d), is there any spatial correlation between apoptotic neurons and activated glial cells? Is there increased proliferation of astrocytes and microglia?
3. The authors show that administration of aSyn fibrils compromised Brain-Chip blood-brain barrier integrity. Is this a direct effect due to fibrils or linked to neurodegeneration/neuroinflammation? Upon fibril administration, can aSyn be detected in the vascular side? This should be investigated in much more depth since this is actually the main question here and one major point of actually using this kind of Brain-Chip.
4. In line with the previous comment, the rescue of Brain-Chip blood brain barrier integrity in Figure 7b by trehalose is important. However, the authors should address experimentally whether the compound could cross the Brain-Chip blood-brain barrier, and specifically protected the endothelial cells or somehow attenuated pathology on the brain side. The rationale for using trehalose needs to be further elaborated.

Minor concerns

- The authors have used iPSC-derived dopaminergic neurons and endothelial cells, yet the other cells are primary human cells. Such an experimental setup complicates the study of cell-to-cell interactions due to variable maturation stages for the different cell types and highly variable genetic background. An isogenic iPSC-based Brain-Chip would be preferable, and this aspect should at least be discussed. In general, the limitations of the present model should be more clearly discussed.
- Was the brain side medium suitable for the survival of all cell types? Accordingly, was the cellular composition comparable between seeding day and day of experiment completion? How was cell number of each cell line controlled? How were the experiments normalized?

- In Supplemental Figure 1, how does dopamine release compare between Brain-Chip and conventional iPSC-derived dopaminergic neuron monoculture?
- Supplemental Figure 3 shows increased no. GFAP+ and CD68+ cells per field, suggesting more astrocytes and microglia, respectively. Was this normalized to the total number of cells?
- The Brain-Chip transcriptomic analysis upon fibril administration is very interesting, but at least some targets including MAO and LRRK2 should be further validated by qRT-PCR.
- The use of the word “develop” confuses the reader in terms of the specific novelties in this study. For instance, both the Brain-Chip device (e.g. lines 74-75), and aSyn fibril paradigm (e.g. lines 81-82) of aSyn anatomical spread have been reported. This should be clarified throughout the text.
- In line with the major concerns, several conclusions must be toned down. Some examples include the claim that the human substantia nigra expression pattern is recapitulated in the Brain-Chip (lines 186-187), or that the data suggests a progressive neurodegeneration and strong neuroinflammatory phenotype (lines 253-254).
- The text should be revised throughout the manuscript, and all abbreviations should be used consistently (e.g. iPS vs iPSC).

Reviewer #2 (Remarks to the Author):

This paper uses organs on chips technology to generate a 3D model of midbrain. Specifically commercially available human dopaminergic neurons are mixed with commercially available cortical human astrocytes, human microglia and pericytes are enshethend by iPSC derived endothelial cells and cultured under fluid flow in microfluidic chambers. Alpha-synuclein fibrils are added and the authors investigate PD phenotypes. They claim p a-syn, mitochondrial impairments, neuroinflammation and compromised barrier function.

While it is interesting to think about new human models for understanding PD, this model consists of 5 different sources of cells from different individuals. The reliability of this model may be questioned due to the mix of individuals and raises major questions and concerns:

What is the physiologic relevance of these experiments? How do the authors rule out interactions between these different sources? How were these different cell types genetically analyzed? What is known about PD relevant genes and SNPs within these donors? What clinical condition is exactly is modeled here? How do alpha-synuclein amounts reflect clinical findings?

The authors claim, that the gene expression analysis were compared to conventional cell cultures. The specific composition of “conventional cell cultures” is not described. This limits the information of the study. A good comparison could be to generate an assembloid of autologous iPSC derived midbrain organoids, astrocytes, microglia, pericytes and endothelial cells and put these under fluid flow.

The depth of analysis of alpha-synuclein pathology is very superficial. E.g. Western blot should be used to further characterize pathological a-syn species (p-syn). CC3 stainings are very variable and require additional cell death assays. Astrocyte and microglia activation markers (GFAP, DC68) should be complemented by extensive expression analysis and add a classical neuroinflammatory stimulus for comparison.

Reviewer #3 (Remarks to the Author):

This manuscript leveraged Organs-on Chips technology to engineer a human Brain Chip representative of the substantia nigra area of the brain, which is most affected in Parkinson's disease. The a-syn fibril model was capable of reproducing key aspects of Parkinson's including phosphorylated a-syn, mitochondrial impairment, neuroinflammation and compromised barrier function. Therefore is proposed to serve as a platform for novel therapeutic interventions.

This manuscript builds on the already published work from this group where they developed a human disease model of drug toxicity-induced pulmonary edema in a lung on a chip microdevice.

This is a very interesting model system which could be potentially very useful to the Parkinson's

community and the field as a whole if the issues below are addressed:

Further information on some of the cells used should be provided: the authors mention primary human astrocytes and microglia, there is no information on the age of the donor or the brain region from which the microglia came from. Were these collected from elective abortions? Were these cells expanded in culture prior to being used in experiments? This information would be useful in determining how akin this set up should be to the adult midbrain which is affected in Parkinson's.

The authors should confirm midbrain dopamine neuron identity in Figure 1. They state on page 5, line 106, that 'First (D0) we seeded in the brain channel human iPS-derived DA neurons' so how many days post neutralisation were these cells? I note in the reporting summary that midbrain markers were used but its not clear if this is by the company who sold them or by the authors themselves.

In the methods there are medias defined as DeSR2 and DeSR1, what are these exactly?

Figure 5a, CC3 staining would appear to be more widespread than the level detailed in the graph on the right, authors should consider using separate channel images to show this more clearly.

On page 21, line 459 there is a reference to Chips were functionalised using Emulate's proprietary reagents. The whole idea of publishing is so that other researchers can benefit by increased scientific knowledge within the scientific community. However, nobody could try to replicate these results without the knowledge of such reagents? This is really not within the spirit of publishing scientific research.

Page 21, line 471 BMEC were seeded at 16-20 million cells per ml. This is a lot of cells and authors should comment on whether this is a rate limiting factor for using this technology?

Responses to Reviewers

We thank the Reviewers for their insightful reviews, the appraisal of the relevance and potential impact of our study in PD research. Their constructive comments helped us to improve the quality and presentation of our work as delineated in the updated manuscript. In the revised manuscript we include a lot of new data to address all your comments as detailed point-by-point below.

Reviewer #1 (Remarks to the Author)

Pediaditakis et al. established a co-culture system modelling the human substantia nigra neurovascular interface, using Brain-Chip technology. Transcriptomic profile comparison suggested that Brain-Chip cultures achieved a more mature state as compared to conventional cultures. Moreover, Brain-Chip and adult substantia nigra showed common differentially expressed genes, as compared to conventional cultures. Exogenous aSyn fibril administration was used as a paradigm for validating Brain-Chip suitability for modelling Parkinson's disease pathomechanisms, and addressing effects on endothelial integrity. In addition to the expected phenotypes of phosphorylated aSyn accumulation, increased overall oxidative stress and neuroinflammation, Brain-Chip blood-brain barrier integrity was compromised by fibrils, and this could be rescued by trehalose. The described co-culture system is potentially highly relevant for studying Parkinson's disease biology, and can have translational value.

Nevertheless, particularly the aspects of Brain-Chip suitability for studying dopaminergic vulnerability and neuron-glia interactions was not sufficiently studied in depth and requires elucidation.

Major concerns

1. Based on Figure 2d, the authors claim that the Brain-Chip recapitulates the gene expression pattern of adult substantia nigra. However, this remains to be addressed via direct Brain-Chip vs. adult substantia nigra RNA-sequencing profile comparison, or more suitable bioinformatics analyses including the 3 groups (Brain-Chip, adult substantia nigra, conventional cultures). Moreover, the global transcriptomic analysis is an important dataset of the study, but in order for the reader to interpret, the conventional cell culture should be defined and described in detail in the manuscript and materials and methods.

2. It is important to clarify whether the Brain-Chip is an advantageous setup for studying dopaminergic neuron vulnerability, and neuron-glia interactions in the paradigm of exogenous aSyn fibril administration. Specifically, Figure 5a-b, shows an overall decrease in neuronal density and increased caspase-3 activation, but this should be quantified for TH+ and TH- neurons. Moreover, the relationship between neuronal loss and neuroinflammation within the Brain-Chip should be elucidated. Which cells of the brain side show uptake of aSyn fibrils at earlier vs. later time points? With regards to glial responses (Figure 5c/d and Supplementary Figure 3c/d), is there any spatial correlation between apoptotic neurons and activated glial cells? Is there increased proliferation of astrocytes and microglia?

3. The authors show that administration of aSyn fibrils compromised Brain-Chip blood-brain barrier integrity. Is this a direct effect due to fibrils or linked to neurodegeneration/neuroinflammation? Upon fibril administration, can aSyn be detected in the vascular side? This should be investigated in much more depth since this is actually the main question here and one major point of actually using this kind of Brain-Chip.

4. In line with the previous comment, the rescue of Brain-Chip blood brain barrier integrity in Figure 7b by trehalose is important. However, the authors should address experimentally whether the compound could cross the Brain-Chip blood-brain barrier, and specifically protected the endothelial cells or somehow attenuated pathology on the brain side. The rationale for using trehalose needs to be further elaborated.

Minor concerns

- The authors have used iPSC-derived dopaminergic neurons and endothelial cells, yet the other cells are primary human cells. Such an experimental setup complicates the study of cell-to-cell interactions due to variable maturation stages for the different cell types and highly variable genetic background. An isogenic iPSC-based Brain-Chip would be preferable, and this aspect should at least be discussed. In general, the limitations of the present model should be more clearly discussed.
- Was the brain side medium suitable for the survival of all cell types? Accordingly, was the cellular composition comparable between seeding day and day of experiment completion? How was cell number of each cell line controlled? How were the experiments normalized?
- In Supplemental Figure 1, how does dopamine release compare between Brain-Chip and conventional iPSC-derived dopaminergic neuron monoculture?
- Supplemental Figure 3 shows increased no. GFAP+ and CD68+ cells per field, suggesting more astrocytes and microglia, respectively. Was this normalized to the total number of cells?
- The Brain-Chip transcriptomic analysis upon fibril administration is very interesting, but at least some targets including MAO and LRRK2 should be further validated by qRT-PCR.
- The use of the word “develop” confuses the reader in terms of the specific novelties in this study. For instance, both the Brain-Chip device (e.g. lines 74-75), and aSyn fibril paradigm (e.g. lines 81-82) of aSyn anatomical spread have been reported. This should be clarified throughout the text.
- In line with the major concerns, several conclusions must be toned down. Some examples include the claim that the human substantia nigra expression pattern is recapitulated in the Brain-Chip (lines 186-187), or that the data suggests a progressive neurodegeneration and strong neuroinflammatory phenotype (lines 253-254).
- The text should be revised throughout the manuscript, and all abbreviations should be used consistently (e.g. iPS vs iPSC).

Responses to Reviewer #1

Thank you for your thorough review. We have performed a number of new experiments to address your specific comments so that the revised manuscript provides a more comprehensive characterization of the effects of α Syn fibrils in brain endothelial cells and of the effects of trehalose including protection of the BBB disruption. In particular, we experimentally showed the following:

- 1) The Brain-Chip can better recapitulate the gene expression pattern of the adult substantia nigra (i.e., the smaller number of statistically significant transcriptomic differences for key signature genes).
- 2) The vulnerability of dopaminergic neurons to α Syn fibrils.
- 3) Detailed characterization of the glial responses and spatial correlation between apoptotic neurons and activated glial cells.
- 4) The contribution of α Syn fibrils to endothelial dysfunction in the Brain-Chip.
- 5) The anti-aggregation and anti-inflammatory actions of trehalose in both the brain and vascular channels.

Please see below our point-by-point response to your comments (underlined text below) including all the data from the new corresponding experiments:

Comment 1. Based on Figure 2d, the authors claim that the Brain-Chip recapitulates the gene expression pattern of adult substantia nigra. However, this remains to be addressed via direct Brain-Chip vs. adult substantia nigra RNA-sequencing profile comparison, or more suitable bioinformatics analyses including the 3 groups (Brain-Chip, adult substantia nigra, conventional cultures). Moreover, the global transcriptomic analysis is an important dataset of the study, but in order for the reader to interpret, the conventional cell culture should be defined and described in detail in the manuscript and materials and methods.

Response: Thank you for this insightful comment that gives us the opportunity to provide additional details in the revised manuscript. **Figure 2d**, depicts a Venn diagram formed by the sets of Differentially Expressed (DE) genes identified after performing differential gene expression (DGE) analysis in each of the following groups: i) Substantia Nigra Brain-Chip vs. Conventional Cell Culture and ii) Adult Substantia Nigra vs. Conventional Cell Culture System. The intersection of the two circles (**Figure 2d**) contains the 209 genes with significantly different expression in the Brain-Chip and in adult substantia nigra, as compared to the conventional cell culture system. We used these 209 DE genes, to perform GO terms enrichment analysis and we identified important biological processes that are statistically significantly enriched in Brain-Chip and adult substantia nigra compared to the conventional cell culture system. The results of the GO terms enrichment analysis are provided as supplementary material (**Supplementary Fig. 1c**).

Following your suggestion, we have also performed direct comparison of the RNA sequencing profiles by performing comparison DGE analyses between the following groups: i) Adult Substantia Nigra vs. Brain-Chip and ii) adult Substantia Nigra vs. Conventional Cell Culture system. The first analysis revealed 566 significantly DE genes in adult substantia nigra, 186 up- and 380 down- regulated. The second analysis revealed

682 DE genes in adult substantia nigra 205 up- and 477 down- regulated. Notably, for both DGE analyses we used the same thresholds to determine the DE genes (adjusted p-value <0.05 and $|\log_2\text{FoldChange}|>1$). The above, show a smaller number of DE genes between adult substantia nigra and Brain-Chips versus the former and conventional culture, suggesting that the Brain-Chip captures more faithfully the gene expression pattern of the adult substantia nigra (i.e., smaller number of statistically significant transcriptomic differences).

To provide further evidence that the Brain-Chip could capture the expression pattern of the adult substantia nigra better from the conventional cell culture system, we identified a list of 13 genes, characteristic of the specific cell types used to build our model (**Fig. 2e**). For each one of these 13 genes, we compared fold change in expression levels in the following groups: i) Adult Substantia Nigra vs. Brain-Chip and ii) Adult Substantia Nigra vs. Conventional Cell Culture system. Our results show that the expression levels of all these genes in the Brain-Chip are more similar to those in the adult substantia nigra, as compared to the conventional cell culture systems (smaller $|\text{Log}_2(\text{Fold Change})|$ values). Notably, for most of these genes (as indicated with asterisks in **Fig. 2e**) the expression changes in the Brain-Chip were significantly smaller as compared to those of the conventional cell culture system.

These results indicate that the gene expression patterns that characterize the primary cell types of the substantia nigra brain tissue are better recapitulated in our Brain-Chip model than in conventional culture systems. In the revised manuscript, we have added a paragraph to discuss all these new findings.

Added text:

On page 9, line 203-To provide further evidence that the Brain-Chip could capture the expression pattern of the adult substantia nigra better from the conventional cell culture system, we identified a list of 13 genes, characteristic of the specific cell types used to build our model (**Fig. 2e**). For each one of these 13 genes, we compared fold change in expression levels in the following groups: i) Adult Substantia Nigra vs. Brain-Chip and ii) Adult Substantia Nigra vs. Conventional Cell Culture system. Our results show that the expression levels of all these genes in the Brain-Chip are more similar to those in the adult substantia nigra, as compared to the conventional cell culture systems (smaller $|\text{Log}_2(\text{Fold Change})|$ values). Notably, for most of these genes (as indicated with asterisks in **Fig. 2e**) the expression changes in the Brain-Chip were significantly smaller as compared to those of the conventional cell culture system.

On page 51, line 1113-Figure 2. e) Gene expression pattern analysis between the following groups: i) Adult Substantia Nigra vs. Brain-Chip and ii) Adult Substantia Nigra vs. Conventional Cell Culture system. In the Brain-Chip, most of these genes (those with an asterisk) exhibit statistically significantly smaller fold changes ($p < 0.01$) in their expression levels as compared to the conventional cell culture system.

The conventional cell cultures and the Brain-Chips were seeded using the same ECM composition as well as cell composition and seeding density. At the first experimental day (D0) the dopaminergic neurons, astrocytes, microglia, and pericytes were seeded on the apical side, followed by the seeding of the endothelial cells (D1) on the basolateral side of the 0.47 cm² Transwell-Clear permeable inserts (0.4- μ m pore size). For the apical compartment we used iCell DopaNeurons Media, while for the basolateral compartment we used HBMEC medium. The cells maintained under static conditions throughout the duration of the experiment (D8). The culture medium was replaced daily in both compartments. This information has been added in the Methods section of the revised manuscript.

Comment 2. It is important to clarify whether the Brain-Chip is an advantageous setup for studying dopaminergic neuron vulnerability, and neuron-glia interactions in the paradigm of exogenous aSyn fibril administration. Specifically, Figure 5a-b, shows an overall decrease in neuronal density and increased caspase-3 activation, but this should be quantified for TH+ and TH- neurons. Moreover, the relationship between neuronal loss and neuroinflammation within the Brain-Chip should be elucidated. Which cells of the brain side show uptake of aSyn fibrils at earlier vs. later time points? With regards to glial responses (Figure 5c/d and Supplementary Figure 3c/d), is there any spatial correlation

between apoptotic neurons and activated glial cells? Is there increased proliferation of astrocytes and microglia?

Response: We agree with the reviewer. A major hallmark of Parkinson's disease is loss of dopaminergic neurons in the substantia nigra pars compacta. To characterize the vulnerability of the dopaminergic neurons in our model of α Syn-induced pathogenesis in the Brain-Chip, we quantified the caspase-3 activation among TH-positive neurons with immunocytochemistry (ICC). These results have been added in the revised manuscript (Results section) as below.

Added text and Images:

On page 12, line 271-Confocal immunocytochemical analysis using antibodies against microtubule-associated protein 2 (MAP2), tyrosine hydroxylase (TH), and cleaved caspase-3 (CC3), confirmed the increase in caspase 3-positive dopaminergic neurons at day six post exposure to α Syn fibrils, as compared to those exposed to monomeric α Syn (Fig. 5a, b).

On page 56, line 1182-Figure 5. a) Representative merged images showing immunostaining for MAP2 (grey), tyrosine hydroxylase (red, TH), and Cleaved Caspase-3 (green, CC3) in the brain channel at six-days post-exposure to α Syn fibrils or α Syn monomers. Scale bars: 100 μ m. b) Quantitation of the number of CC3 and MAP2- or TH-positive neurons. Statistical analysis by two-way ANOVA with Tukey's multiple comparisons test (3~4 randomly selected different areas per chip, n=3 independent chips/experimental group, **p<0.01, ****p<0.0001, NS: Not Significant). Error bars represent mean \pm SEM. Scale bars: 100 μ m.

Response: To address this comment, we assessed the cellular uptake of α Syn fibrils in the brain channel, site of the co-culture of all brain cell types in the chip, at day three and six post exposure to α Syn fibrils by measuring the induction of phospho- α Syn129. Dual immunofluorescent staining for phospho- α Syn129 and specific markers for each cell type in the culture, was used to characterize the expression of phospho- α Syn129 at the different timepoints. All these new results have been added in the revised manuscript (Results and Supplementary Information), as below.

Added text and images:

On page 11, line 234-Immunofluorescence staining showed phospho- α Syn129 accumulation at three- and six-days post exposure to α Syn fibrils within TH positive neurons (**Supplementary Fig. 3b, and Fig. 3e**), astrocytes and microglia (**Supplementary Fig, 3c,d and Fig. 3f,g**), in agreement with previous reports (Courte et al., 2020).

On page 53, line 1136-Figure 3. e) Confocal images of double immunostaining, for phospho- α Syn129 (green) and TH (red), in the brain channel at day six post-exposure (D8) to α Syn fibrils. Scale bars: 50 μ m. f, g) α Syn fibrils are taken up by astrocytes and microglia, as evidenced by double immunostaining for phospho- α Syn129 (green) and either GFAP (red) for astrocytes or IBA1 (red) for microglia in the brain channel at day six post-exposure (D8) to α Syn fibrils. Scale bars: 50 μ m.

(Supplementary Information) **On page 5, line 66-Supplementary Figure 3.** b) Confocal images of double immunostaining for phospho- α Syn129 (green) and TH (red) in the brain channel after three days upon-exposure to α Syn fibrils (D5 of the experiment). Scale bars: 50 μ m. c, d) α Syn fibrils are taken up by astrocytes and microglia after three days upon-exposure to α Syn fibrils (D5 of the experiment); double immunostaining for phospho- α Syn129 (green) and either GFAP (red) for astrocytes or IBA1 (red) for microglia. Scale bars: 50 μ m.

Response: We thank the reviewer for the insightful comment. To this end, we tested the spatial correlation between apoptotic neurons (TUNEL-positive) and activated microglia (CD68-positive) at day six post exposure to α Syn fibrils or monomeric α Syn. Our results

support the notion that activated microglia act as a sensitive index of neuropathological changes (Imamura et al., 2003).

Next, we sought to assess if it was the proliferation of glia, or its higher immunoreactivity driving the observed glia responses to α Syn in the Brain-Chip. Our data, based on evaluation of the absolute cell numbers and the expression of relevant specific markers, suggest that GFAP+ astrocytes and CD68+ activated microglia were significantly increased in the α Syn-induced model (see also **Comment 11**). No detectable change was found in the total number of cells. We also assessed the glial cell proliferation using the Ki67 staining nuclear protein and found that α Syn fibrils promoted glial activation but not its proliferation, as compared to exposure to monomers, in line with other studies (Harm et al., 2017). Thus, we believe that the observed astrocytic and microglial changes, represent mainly the reaction/response of the existing resting glial cells, rather than the generation of new cells by glial precursors. These new results have been included in the Supplementary Information.

Added text and images:

On page 13, line 288-Interestingly, a large number of activated microglia formed clusters around the apoptotic neurons (**Supplementary Fig. 5d**), suggesting that the microglia response may be linked to the neuronal injury.

(Supplementary Information) **On page 9, line 106-Supplementary Figure 5. d)** Representative confocal images showing immunostaining for MAP2 (gray), TUNEL (green), and CD68 (red) in the brain channel at day six post exposure to α Syn fibrils or α Syn monomers. Scale bars: 100 μ m.

On page 13, line 290-Finally, we investigate the effect α Syn fibrils on the proliferation of glial cells. Immunocytochemical staining with the proliferation marker Ki67 showed no significant increase in proliferation following exposure to α Syn fibrils, but not to monomeric α Syn (**Supplementary Fig. 5e, f**). In line, there were no differences in the number of cells between the control and treated groups (**Supplementary Fig. 5g**). Thus, we conclude that the astrocytic and microglial responses to α Syn fibrils in our model are

due largely due to changes in the existing resting glial cells, rather than due to the generation of new glial cells from glial precursors.

Although adding a classical neuroinflammatory stimulus (TNF- α ; 100ng/mL) validate our results, we would like to kindly request the TNF- α treated group not to be included in the final figure since it is a preliminary result of an ongoing study.

On page 9, line 108-Supplementary Figure 5. e) Immunocytochemical staining of proliferating Ki67-positive cells in α Syn monomeric or α Syn fibril-treated cultures. Scale bars: 100 μ m. f) Percentage of the Ki67-positive cells normalized to the total number of cells. Statistical analysis by Student's t test (4 randomly selected different areas per chip, n=3 independent chips/experimental group, NS: Not Significant). Error bars represent mean \pm SEM. Scale bars: 100 μ m. g) The nuclei count quantified from DAPI staining remained similar between the control and treated groups. Statistical analysis by Student's t test (n=3 Brain-Chips with 4 randomly selected different areas per chip, NS: Not Significant). Error bars represent mean \pm SEM.

Comment 3. The authors show that administration of aSyn fibrils compromised Brain-Chip blood-brain barrier integrity. Is this a direct effect due to fibrils or linked to neurodegeneration/neuroinflammation? Upon fibril administration, can aSyn be detected in the vascular side? This should be investigated in much more depth since this is actually the main question here and one major point of actually using this kind of Brain-Chip.

Response: We thank the reviewer for this important comment. Indeed, data included in the revised manuscript support the hypothesis of endothelial pathology as an important contributor in the pathophysiology of PD. In particular, we show compromised blood-brain barrier integrity, activation of ICAM-1 (intercellular adhesion molecule-1) shown to be upregulated in endothelial cells in inflammatory conditions (Marchetti & Engelhardt, 2020), and increase of the secreted proinflammatory cytokines in the vascular channel (discussed in **Comment 7** below). In the revised manuscript we also include new data showing accumulation of phosphorylated α Syn (Ser129) in the endothelial cells. Previous studies have linked α -synuclein to the mitochondrial protein import machinery, by its binding to TOM20 (translocase of the outer membrane) and the corresponding impact on the mitochondrial protein import (De Miranda et al. 2020). We identified decreased levels of TOM20 protein levels in endothelial cells following exposure to α Syn fibrils, but not to the monomeric α Syn, in further support of the effect of α Syn fibrils in the brain endothelial cells. These new results are included in the revised manuscript (Result) and Supplementary Information as below.

Added text and images:

On page 14, line 316-We next assessed whether increased BBB permeability was associated to disruption of the tight junction protein ZO-1. Staining of the endothelial cells in the vascular channel of the Brain Chip with anti-ZO1 at D8, demonstrated tight junctions damage following exposure to α Syn fibrils, but not to monomeric α Syn (**Supplementary Fig. 6b,c**). Next, we examined the implication of mechanisms attributable to BBB impairment, such as changes in the expression of TOM20, which prevents α -synuclein-induced mitochondrial dysfunction (De Miranda et al. 2020), and ICAM-1, a key molecule in immune-mediated and inflammatory processes in endothelial cells (Marchetti & Engelhardt, 2020). Immunofluorescence analysis showed attenuated expression of the TOM20 and increased expression of ICAM-1 at six days (experimental D8) post exposure to α Syn fibrils compared to monomeric group (**Supplementary Fig. 6b,c**), suggesting a possible role for α Syn fibrils in BBB damage correlated with mitochondria dysfunction and inflammation in the vasculature. To check whether α Syn fibrils were directly deposited on the endothelial cells in the Brain Chip, we performed immunofluorescent staining with phospho- α Syn129 in the vascular channel. Our results show that exposure of the Substantia Nigra Brain-Chip to α Syn fibrils induced phosphorylation of α Syn localized in the endothelial cells in the brain channel, six days post exposure to α Syn fibrils (**Fig. 6c,d**). These results indicate that α Syn fibrils affect the permeability of the BBB in the Brain-Chip, although our experiment cannot distinguish between a direct role of α Syn fibrils- or indirect via effects on other contributing molecules in causing BBB impairment.

(Supplementary Information) **On page 10, line 127-Supplementary Figure 6.** b) Representative merged image of immunofluorescent staining of tight junction protein 1 (ZO-1, red), Mitochondrial import receptor subunit (TOM20, green), and cell nuclei (DAPI, blue). Scale bars: 100 μm . c) Representative merged image of immunofluorescent staining of tight junction protein 1 (ZO-1, red), Intercellular Adhesion Molecule 1 (ICAM-1, green), and cell nuclei (DAPI, blue). Scale bars: 100 μm .

On page 59, line 1216-Figure 6. c) Immunofluorescence micrographs show accumulation of phosphorylated αSyn (green, phospho- $\alpha\text{Syn}129$ staining; red, ZO-1; blue, DAPI) in the vascular channel of the Brain Chip, following treatment with αSyn fibrils, in contrast to lack of detected pathology in response to αSyn monomers. Scale bars: 100 μm . d) Quantitative analysis of fluorescence intensity in each group at day six post exposure to αSyn fibrils or αSyn monomers. Statistical analysis by Student's t test ($n=4$ independent chips with 3~5 randomly selected different areas per chip, **** $P<0.0001$ compared to monomeric group). Error bars represent mean \pm SEM.

Comment 4. In line with the previous comment, the rescue of Brain-Chip blood brain barrier integrity in Figure 7b by trehalose is important. However, the authors should address experimentally whether the compound could cross the Brain-Chip blood-brain barrier, and specifically protected the endothelial cells or somehow attenuated pathology on the brain side. The rationale for using trehalose needs to be further elaborated.

Response: We thank the reviewer for his/her comment. Trehalose, an autophagy stimulating disaccharide, is a BBB-permeable molecule that has been shown to prevent/ameliorate protein(s) fibrillation and formation of aggregated deposits in the brain in experimental models of neurodegenerative diseases (Debnath et al., 2017; Khalifeh et al., 2019, 2020; Pagano et al., 2020). The study of Hoffmann et al. provided evidence that trehalose prevents or halts the propagation of α Syn pathology by targeting lysosomes. Modulation of inflammatory responses by trehalose has been also reported (Echigo et al., 2012). Studies in aged mice showed improvement of vascular autophagy and age related endothelial dysfunction after oral supplementation with trehalose (LaRocca et al., 2012). In the previously submitted manuscript we have shown that trehalose significantly decreases BBB permeability and rescues damaged tight junctions following exposure to α Syn fibril.

In the revised manuscript, we performed additional experiments and confirmed that trehalose crosses the BBB in the Brain-Chip and prevents lysosomal alterations within α Syn-exposed cells in the brain channel. In addition, trehalose attenuates α Syn accumulation and inflammatory responses in both the brain and vascular channels of the chip. Notably, this is the first study testing trehalose in a human cell-based model of the neurovascular unit. All the new results have been added in the revised manuscript, "Results" and Supplementary Information as shown below.

Added text and images:

On page 17, line 382-To evaluate whether trehalose crosses the BBB and attenuates α Syn accumulation within α Syn-exposed cells, we first measured trehalose penetration in the brain channel. Using the trehalose assay kit, we found that more than 60% of trehalose had cross the BBB (**Fig. 7b**). As accumulation of phosphorylated α Syn in fibril-exposed cells was readily detectable in the brain channel (**Fig. 3**), we next validated the efficacy of trehalose in preventing lysosomal alteration and accumulation of aggregated α Syn. In agreement with previous findings (Hoffmann et al., 2019), lysosomal cathepsin D activity decreased significantly in α Syn fibrils exposed cells, indicative of impaired lysosomal activity (**Supplementary Fig. 7a**). Our results showed that trehalose prevented lysosomal alterations within α Syn-exposed cells in the brain channel (**Supplementary Fig. 7a**) and significantly decreased the abundance of phospho- α Syn129 compared to the non-treated chips (**Fig. 7c and Supplementary Fig. 7b**). Taken together, these findings demonstrate the protective effects of trehalose on the α Syn-mediated impairment of lysosomes and on the clearance of accumulated α Syn in the cells within the brain channel. Next, we examined whether trehalose suppressed the secretion of proinflammatory cytokines detected in α Syn fibrils-exposed cells in the brain channel (**Fig. 5e,f**). We found that the TNF- α and IL-6 levels were significantly decreased in the brain channel's effluent following exposure to trehalose compared to the non-treated chips, in support of its protective effects against neuroinflammation (**Fig. 7d**).

At the vascular channel, trehalose exerted similar effects as it reduced the levels of phospho- α Syn129 (**Fig. 7e and Supplementary Fig. 7c**) and the secreted IL-6 and INF- γ (**Fig. 7f**). INF- γ levels were found significantly increased, following exposure to α Syn fibrils versus monomeric α Syn, only in the effluent of the vascular channel (**Fig. 7d,f**). Of note, IFN- γ levels are elevated in the blood plasma of PD patients (Mount et al., 2007) and reportedly increase BBB permeability (Chrobak et al., 2012; Ng et al., 2015). These

findings revealed a distinct, channel-specific profile of secreted proinflammatory cytokines in the α Syn fibrils-induced on chip-model of the disease, of potential significance for PD pathogenesis and targeting of the corresponding neuroinflammation.

On page 61, line 1256-Figure 7. Effect of autophagic inducer, trehalose on the α Syn induced model. a) Schematic of the experimental design for the perfusion of trehalose and α Syn fibril in the Brain-Chip b) Percentage of trehalose crossing the BBB after 24 hrs of perfusion. c) Quantitation of the accumulated phosphorylated α Syn at the brain channel based on the fluorescence intensity at day 8 in the α Syn fibril model with or without trehalose treatment. Statistical analysis by two-way ANOVA with Tukey's multiple comparisons test ($n=3\sim 4$ independent chips with $3\sim 5$ randomly selected different areas per chip, $***P<0.001$, $****P<0.0001$). Error bars represent mean \pm SEM. d) Secreted levels of TNF- α , IL-6, and INF- γ at the brain channel at D8. Statistical analysis by two-way ANOVA with Tukey's multiple comparisons test ($n=4$ independent chips with $3\sim 5$ randomly selected different areas per chip, $**P<0.01$, $***P<0.001$, $****P<0.0001$, NS: Not Significant). Error bars represent mean \pm SEM. e) Quantitation of the accumulated phosphorylated α Syn at the vascular channel based on the fluorescence intensity at day

8 in the α Syn fibril model with or without trehalose treatment. Statistical analysis by two-way ANOVA with Tukey's multiple comparisons test ($n=4$ independent chips with 3~5 randomly selected different areas per chip, **** $P<0.0001$). Error bars represent mean \pm SEM. f) Secreted levels of TNF- α , IL-6, and INF- γ at the vascular channel at day 8 in the α Syn fibril model with or without trehalose treatment. Statistical analysis is two-way ANOVA with Tukey's multiple comparisons test ($n=3\sim 4$ independent chips with 3~5 randomly selected different areas per chip, ** $P<0.01$, *** $P<0.001$, NS: Not Significant). Error bars represent mean \pm SEM.

(Supplementary Information) **On page 13, line173-Supplementary Figure 7.** Effect of trehalose on the accumulation of phosphorylated α Syn. a) Lysosomal cathepsin D activity at the brain channel on day 8. Statistical analysis by two-way ANOVA with Tukey's

multiple comparisons test (n=3 independent chips with duplicate technical replicates assayed per condition). Error bars represent mean±SEM. b) Immunofluorescence micrographs depict the accumulation of phosphorylated αSyn (green, phospho-αSyn129 staining; blue, DAPI) at day 8 in the αSyn fibril model with or without trehalose treatment. Absence of pathology in the brain channel following exposure to monomer or co-treatment with fibrils and trehalose. c) Immunofluorescence micrographs depict the effect of fibrils on the accumulation of phosphorylated αSyn and tight junctions (green, phospho-αSyn129 staining; red, ZO-1; blue, DAPI) at day 8 in the αSyn fibril model with or without trehalose treatment. Absence of pathology in the brain channel following exposure to monomer or co-treatment with fibrils and trehalose. Scale bars: 100 μm.

Comment 5. The authors have used iPSC-derived dopaminergic neurons and endothelial cells, yet the other cells are primary human cells. Such an experimental setup complicates the study of cell-to-cell interactions due to variable maturation stages for the different cell types and highly variable genetic background. An isogenic iPSC-based Brain-Chip would be preferable, and this aspect should at least be discussed. In general, the limitations of the present model should be more clearly discussed.

Response: We agree with the Reviewer that the ideal Brain-Chip would be an isogenic model with all cell types originating from the same human individual, enabling most precise characterization of the cell-cell interactions and personalized medicine applications. However, an isogenic iPSC-based Brain-Chip requires robust and reproducible differentiation protocols for derivation of the several cell types of the CNS, validated and optimized by different groups. Even though the progress is tremendous, and some of the required differentiation protocols already showed high robustness, the field has not yet reached the point where a reproducible, standardized isogenic model is available for disease modeling studies (Rivetti di Val Cervo et al., 2021). Variability between models remains a considerable issue, as exemplified in a study comparing the differentiation capacity of four different iPSC lines to isogenic BBB models, including endothelial cells and astrocytes (Patel et al., 2017). While efforts continue to address the need for well-defined and characterized mature cell types, employed models based on primary human cells, albeit of mixed origin, provide new significant insights in our understanding of neurodegenerative diseases. A recent study successfully incorporated isogenic microglia-like cells with brain region-specific organoids from hiPSCs, however these structures lacked the endothelial component required for BBB formation (Song et al., 2019). Therefore, for this study we chose to employ well-characterized and established cell sources to develop stable and standardized assays that provide good cell quality and enable comparisons between experimental groups based on clinically relevant endpoints. We have reedited the Discussion to address this very important point in the revised manuscript (Discussion).

Comment 6. Was the brain side medium suitable for the survival of all cell types? Accordingly, was the cellular composition comparable between seeding day and day of experiment completion? How was cell number of each cell line controlled? How were the experiments normalized?

Response: Thank you for the comments. We have conducted additional experiments to demonstrate the suitability of the medium in the brain channel for the survival of all cell types seeded there. We measured the metabolic activity of each cell type cultured in their nominal medium (provided by the vendor) or in the medium we used for the brain channel (“DA neurons medium”) using the MTS assay as described in previous studies (Ahn et al., 2020). Additionally, we assessed the morphological characteristics of the cells maintained in the brain channel medium by bright-field images and immunofluorescence staining. Astrocytes, microglia, and pericytes maintained their phenotypic characteristics without biased overgrowth in the neuronal culture. In line, the cellular composition in the brain channel was sustained throughout the length of culture, i.e., from day 2 when all cells were seeded up to the completion of the experiment on day 8. Staining for cell type-specific markers was done, and quantitative results were normalized to the total number of cells. In particular, the nuclei count as quantified based on DAPI staining remained stable between day 2 and day 8. All these findings and the associated descriptions are included in the revised manuscript (Result, Methods, and the Figure legend).

Added text and images:

On page 6, line 122-We used cell morphology and metabolic activity assays (see Methods for details), to confirm that the culture conditions and the DA neurons medium used, were able to support the cellular composition of the brain channel throughout the length of the study (**Supplementary Fig. 2a-d**).

(Supplementary Information) **On page 3, line 35-Supplementary Figure 2.** Characterization of the human Brain-Chip. a) Morphology of astrocytes, microglia, and pericytes cultured in their respective culture medium or the “brain channel medium”. b) Cellular metabolic activities assessed by a MTS assay to compare astrocyte medium, microglia medium, pericyte medium, and brain channel medium (DA neurons medium) (n=6/group, NS: Not Significant after comparison between brain channel medium versus each cell culture medium by student t-test). c) Immunofluorescent microphotographs (left) validate the dopaminergic neurons with MAP2 (green), astrocytes with GFAP (magenta) and pericytes (red), and the DAPI (blue) for cell nuclei. Immunofluorescent microphotographs (right) of the glia culture: astrocytes (magenta, GFAP staining), and resting microglia (yellow, TMEM119). Scale bars: 50 μ m. d) Relative percentage representation of each cell type in the brain channel throughout the length of culture (day 2, start and day 8, at the completion of the experiment). Staining with cell type specific markers (TH; dopaminergic neurons, GFAP; astrocytes, IBA1; microglia, NG2; pericytes) followed by normalization to the total cell number. Nuclei counts based on DAPI staining were similar in days 2 and 8. (n=6 Brain-Chips; 8 randomly selected different areas per chip). Error bars represent mean \pm SEM.

Comment 7. In Supplemental Figure 1, how does dopamine release compare between Brain-Chip and conventional iPSC-derived dopaminergic neuron monoculture?

Response: We measured the dopamine levels released from Brain-Chip and conventional iPSC-derived dopaminergic neuron monoculture after 2- and 8-days post-seeding.

Added text:

On page 6, line 138-Next, we assessed the secreted dopamine levels via enzyme-linked immunosorbent assay (ELISA) to confirm the functionality of the dopaminergic neurons in the Brain-Chip. As shown (**Supplementary Fig. 2g**), the secretion of dopamine was stable over the length of the culture. Notably, the dopamine levels were significantly higher in the Brain Chip as compared to the monoculture of iPSC-derived dopaminergic neurons, indicative of the effect of a physiologically relevant cellular environment, such as of the Brain-Chip, in the functional maturation of the DA neurons.

(Supplementary Information) **On page 4, line 55-Supplementary Figure 2. g)** ELISA for dopamine secreted into the medium of the brain channel and conventional iPSC-derived dopaminergic neuron monoculture on days 2 and 8. Statistical analysis by Student's t test ($n=3$ independent chips with duplicate technical replicates assayed per condition). Error bars represent mean \pm SEM.

Comment 8. Supplemental Figure 3 shows increased no. GFAP+ and CD68+ cells per field, suggesting more astrocytes and microglia, respectively. Was this normalized to the total number of cells?

Response: Thank you for pointing this out. We normalized our measurement to the total number of cells. Our data suggest that the GFAP+ astrocytes and CD68+ activated microglia, but not the total number of glial cells, is significantly increased in the α Syn fibrils-based disease model. Taken together our response to **Comment 5** above, we conclude that glial responses/activation in the α Syn induced model are largely due to higher protein amounts, rather than cell proliferation (see also Serrano-Pozo et al., 2013; Chung et al., 2019).

Added images:

(Supplementary Information) **On page 8, line 99-Supplementary Figure 5.** b) Quantification of the number of GFAP-positive events per field of view normalized to the total number of cells. Statistical analysis by Student's t test (n=3 independent chips with 4 randomly selected different areas per chip, **p<0.01 compared to monomeric group). c) Quantification of the number of CD68-positive events per field of view normalized to the total number of cells. Statistical analysis by Student's t test (n=3 Brain-Chips with 4 randomly selected different areas per chip, ****P<0.0001 compared to the monomeric group). Error bars present mean \pm SEM.

Comment 9. The Brain-Chip transcriptomic analysis upon fibril administration is very interesting, but at least some targets including MAO and LRRK2 should be further validated by qRT-PCR.

Response: We validated the gene expression of specific targets by qRT-PCR as suggested by the Reviewer. We confirmed the high expression levels of PD-associated genes at the vascular channel supported by our RNA-Seq findings.

Added text:

On page 16, line 355-The altered expression of PD-associated genes was confirmed by quantitative RT-PCR (qPCR) (**Supplementary Fig. 6e**).

(Supplementary Information) **On page 11, line 135-Supplementary Figure 6. e)** Expression of APOA1, SNCAIP, LRRK2, and MAOA in the vascular channel at day six post exposure to α Syn fibrils or α Syn monomers. Statistical analysis by Student's t-test (n=4 independent chips, **p < 0.01, ***p < 0.001, ****p < 0.0001 compared to α Syn monomeric group). Error bars represent mean \pm SEM.

RNA isolation, reverse transcription, and qPCR. Each channel of the Brain-Chip was lysed using TRI Reagent® (Sigma- Aldrich, T9424) and RNA was isolated using the Direct-zol RNA Purification Kit (Zymo Research, R2060). Genomic DNA was removed using the TURBO DNA-free™ Kit (ThermoFischer, AM1907) and reverse transcription to cDNA was performed with the SuperScript IV Synthesis System (ThermoFischer, 18091050). TaqMan Fast Advanced Master Mix (Applied Biosystems, 4444963) and TaqMan Gene Expression Assays were used for the Quantitative Real Time PCR (qPCR). The qPCR was detected on a QuantStudio 3 PCR System (Fisher Scientific). The target genes were assessed using commercially available primers (GFAP: Hs00909233_m1, VIM: Hs00958111_m1, LCN2: Hs01008571_m1, CD68: Hs02836816_g1, EDA: Hs03025596_s1, APOA1: Hs00163641_m1, SNCAIP: Hs00917422_m1, LRRK2: Hs01115057_m1, MAOA: Hs00165140_m1). All primers are also listed in **Supplementary Table 2**. The results were quantified by the comparative Ct method. Ct values for samples were normalized to the expression of the housekeeping gene (18s: Hs99999901_s1; Applied Biosystems).

Comment 10. The use of the word “develop” confuses the reader in terms of the specific novelties in this study. For instance, both the Brain-Chip device (e.g. lines 74-75), and aSyn fibril paradigm (e.g. lines 81-82) of aSyn anatomical spread have been reported. This should be clarified throughout the text.

Response: We agree with the Reviewer that the Brain-Chip has been reported previously and the word “develop” could be confusing. The novelty of the Brain Chip model we have built and present in this study is that it is the first-time dopaminergic neurons and microglia have been used in the Brain-Chip, together with the BBB elements, i.e., endothelial cells, astrocytes, and pericytes previously described. These developments significantly expanded the capabilities of the previously reported multicellular human Brain/Blood-

Brain Barrier (BBB) models and enabled studies closer to disease modeling. For instance, by using a well-studied and acceptable approach to induce PD-relevant phenotypes we were able to get *in vivo* relevant responses in the Brain-Chip and even more to confirm its sensitivity to agents of potential therapeutic value. Based on the reviewer's suggestion, we have corrected all related points in the revised manuscript.

Added text:

On page 4, line 74- In the present study, we describe a novel approach where we have **populated** a Brain-Chip with human dopaminergic neurons a characteristic cell type in the substantia nigra, the area predominantly affected in PD.

On page 4, line 81- In order to model states of exposure to abnormal α Syn aggregation and confirm the capability of the Substantia Nigra Brain-Chip to generate clinically relevant endpoints, we **reconstructed** a model of synucleinopathy by introducing human α Syn pre-formed fibrils (PFFs or "aSyn fibrils") within the brain channel.

On page 5, line 97-Reconstitution and characterization of a human Substantia Nigra Brain-Chip.

On page 20, line 437-We **generated** an engineered human Substantia Nigra Brain-Chip

Comment 11. In line with the major concerns, several conclusions must be toned down. Some examples include the claim that the human substantia nigra expression pattern is recapitulated in the Brain-Chip (lines 186-187), or that the data suggests a progressive neurodegeneration and strong neuroinflammatory phenotype (lines 253-254).

Response: We re-edited the manuscript to address the Reviewer's comments- thank you.

Comment 12. The text should be revised throughout the manuscript, and all abbreviations should be used consistently (e.g. iPS vs iPSC).

Response: Thank you. We corrected all typos in the manuscript and checked for abbreviations consistency.

Reviewer #2 (Remarks to the Author)

This paper uses organs on chips technology to generate a 3D model of midbrain. Specifically commercially available human dopaminergic neurons are mixed with commercially available cortical human astrocytes, human microglia and pericytes are enshethend by iPSC derived endothelial cells and cultured under fluid flow in microfluidic chambers. Alpha-synuclein fibrils are added and the authors investigate PD phenotypes. They claim p a-syn, mitochondrial impairments, neuroinflammation and compromised barrier function.

While it is interesting to think about new human models for understanding PD, this model consists of 5 different sources of cells from different individuals. The reliability of this model may be questioned due to the mix of individuals and raises major questions and concerns:

What is the physiologic relevance of these experiments? How do the authors rule out interactions between these different sources? How were these different cell types genetically analyzed? What is known about PD relevant genes and SNPs within these donors? What clinical condition is exactly is modeled here? How do alpha-synuclein amounts reflect clinical findings?

The authors claim, that the gene expression analysis were compared to conventional cell cultures. The specific composition of "conventional cell cultures" is not described. This limits the information of the study. A good comparison could be to generate an assembloid of autologous iPSC derived midbrain organoids, astrocytes, microglia, pericytes and endothelial cells and put these under fluid flow.

The depth of analysis of alpha-synuclein pathology is very superficial. E.g. Western blot should be used to further characterize pathological a-syn species (p-syn). CC3 stainings are very variable and require additional cell death assays. Astrocyte and microglia activation markers (GFAP, DC68) should be complemented by extensive expression analysis and add a classical neuroinflammatory stimulus for comparison.

Responses to Reviewer #2

We would like to thank the reviewer for her/his insightful comments (underlined in the text below), we have addressed in the revised manuscript, as we have:

- 1) Performed an additional in-depth characterization of the cell sources, including evaluation of PD related genes and SNPs in each donor.
- 2) Confirmed the phosphorylation of α Syn by immunoblot analysis.
- 3) Performed a thorough assessment of dopaminergic neuron vulnerability using an additional cell death assay (TUNEL).
- 4) Performed quantitative analysis on the gene expression of reactive astrocyte and activated microglia markers, we found them upregulated in our α Syn fibrils-induced injury in the Brain-Chip.

Please find below the point-by-point response to each comment:

General Comments. While it is interesting to think about new human models for understanding PD, this model consists of 5 different sources of cells from different individuals. The reliability of this model may be questioned due to the mix of individuals and raises major questions and concerns: What is the physiologic relevance of these experiments? How do the authors rule out interactions between these different sources? How were these different cell types genetically analyzed? What is known about PD relevant genes and SNPs within these donors? What clinical condition is exactly modeled here? How do alpha-synuclein amounts reflect clinical findings?

Response: Thank you for your comments. As discussed in **Comment 8 of Reviewer #1** above, we agree with the Reviewer that the ideal Brain-Chip would be an isogenic model with all cell types originating from the same human individual, enabling most precise characterization of the cell-cell interactions and personalized medicine applications. However, an isogenic iPSC-based Brain-Chip requires robust and reproducible differentiation protocols for derivation of the several cell types of the CNS, validated and optimized by different groups. Even though the progress is tremendous, and some of the required differentiation protocols already showed high robustness, the field has not yet reached the point where a reproducible, standardized isogenic model is available for disease modeling studies (Rivetti di Val Cervo et al., 2021). Variability between models remains a considerable issue, as exemplified in a study comparing the differentiation capacity of four different iPSC lines to isogenic BBB models, including endothelial cells and astrocytes (Patel et al., 2017). While efforts continue to address the need for well-defined and characterized mature cell types, employed models based on primary human cells, albeit of mixed origin, provide new significant insights in our understanding of neurodegenerative diseases. A recent study successfully incorporated isogenic microglia-like cells with brain region-specific organoids from hiPSCs, however these structures lacked the endothelial component required for BBB formation (Song et al., 2019). Therefore, for this study we chose to employ well-characterized and established cell sources to develop stable and standardized assays that provide good cell quality and enable comparisons between experimental groups based on clinically relevant endpoints.

We have reedited the Discussion to address this very important point in the revised manuscript (Discussion).

By leveraging next-generation sequencing data and information retrieved from well-curated databases providing signature gene sets characteristic for the healthy human substantia nigra adult tissue, we were able to identify signatures specific for each cell type included in the Brain Chip. We show that the cells on Brain-Chip exhibit gene expression profiles indicative of their maturation state, and similar to human adult substantia nigra tissue, as compared to that of the conventional culture systems (**Fig. 3e**). To the best of our knowledge (based on data provided by the selling company) these donors do not carry mutations in genes and SNPs that have been associated with PD susceptibility. Further analysis of the transcriptomic analysis and SNPs genotyping (**Supplementary Fig. 1a, b**) of the Brain-Chip, showed expression of PD-related genes (Klein and Westenberger, 2012) at levels similar to those in healthy adult human substantia nigra tissue. We did not identify common LRRK2 polymorphisms in either of the donors.

Added text and images:

On page 5, line 112-To the best of our knowledge (based on data provided by the selling company) these donors do not carry any of the mutations in genes and SNPs associated with PD susceptibility. Further analysis of the transcriptomic profiling and SNPs genotyping of the Brain-Chip, showed expression of PD-related genes (Klein and Westenberger, 2012) at levels similar to those in the healthy adult human substantia nigra tissue. Common LRRK2 polymorphisms, such as rs34637584 and rs33949390, were not detected in any of the donors used (**Supplementary Fig. 1a, b**).

On page 1, line 3-Supplementary Figure 1. Genetic similarities between the Brain-Chip and the normal adult substantia nigra. a) Average gene expression levels (Transcripts Per Million - TPM) of PD related genes in Brain-Chip and normal adult substantia nigra tissue. b) Allelic discrimination plots obtained for rs34637584 and rs33949390 using TaqMan genotyping assay. Dots represent genotypes and the squares on the bottom left of the plot are no-template control.

TaqMan genotyping assay. Genomic DNA was isolated from each cell type using PureLink genomic DNA purification kit (Thermo Fisher Scientific) following the manufacturer's recommended protocol. DNA concentrations were determined using the NanoDrop 2000 UV/VIS spectrophotometer (Thermo Fisher Scientific). Genotyping by

TaqMan was performed following as per manufacturer's instructions using 10 ng of DNA mixed with the TaqMan Genotyping Master Mix (Thermo Fisher Scientific) and custom TaqMan SNP assays (Thermo Fisher Scientific), **Supplementary Table 1**.

Specific Comments:

Comment 1. The authors claim, that the gene expression analysis were compared to conventional cell cultures. The specific composition of "conventional cell cultures" is not described. This limits the information of the study. A good comparison could be to generate an assembloid of autologous iPSC derived midbrain organoids, astrocytes, microglia, pericytes and endothelial cells and put these under fluid flow.

Response: Thank you for your comment. The conventional cell cultures and the Brain-Chips were seeded using the same ECM composition as well as cell composition and seeding density. At the first experimental day (D0) the dopaminergic neurons, astrocytes, microglia, and pericytes were seeded on the apical side, followed by the seeding of the endothelial cells (D1) on the basolateral side of the 0.47 cm Transwell-Clear permeable inserts (0.4- μ m pore size). For the apical compartment we used iCell DopaNeurons Media, while for the basolateral compartment we used HBMEC medium. The cells maintained under static conditions throughout the duration of the experiment (D8). The culture medium was replaced daily in both compartments. This information has been added in the Methods section of the revised manuscript.

Response: Thank you for bringing up this important point.
As discussed in the **General Comments of Reviewer #3** above:

"We agree with the Reviewer that the ideal Brain-Chip would be an isogenic model with all cell types originating from the same human individual, enabling most precise characterization of the cell-cell interactions and personalized medicine applications. However, an isogenic iPSC-based Brain-Chip requires robust and reproducible differentiation protocols for derivation of the several cell types of the CNS, validated and optimized by different groups. Even though the progress is tremendous, and some of the required differentiation protocols already showed high robustness, the field has not yet reached the point where a reproducible, standardized isogenic model is available for disease modeling studies (Rivetti di Val Cervo et al., 2021). Variability between models remains a considerable issue, as exemplified in a study comparing the differentiation capacity of four different iPSC lines to isogenic BBB models, including endothelial cells and astrocytes (Patel et al., 2017). While efforts continue to address the need for well-defined and characterized mature cell types, employed models based on primary human cells, albeit of mixed origin, provide new significant insights in our understanding of neurodegenerative diseases".

Recently, examples on isogenic human models developed by employing iPSC-derived midbrain organoids (Galet et al., 2020). These are very promising developments but not yet suitable for our purposes, as brain organoids in their current stage of development lack the reproducibility required for development of standardized brain platforms, due to

the stochastic generation of neural tissue. The latter impacts on the self-organization and self-development capacities of the tissue and does not prevent random differentiation of the diverse neuronal types in the organoid. Although big efforts by a number of academic and industry groups aim to provide reproducible protocols and supporting Organ-Chip designs for controlled development and use of tissue structures such as intestinal organoids (Kasendra et al. 2020), more extensive engineering and optimization is needed to incorporate brain organoids on chips to recreate the neurovascular unit. A recent study successfully incorporated isogenic microglia-like cells with brain region-specific organoids from hiPSCs, however these structures lacked the endothelial component required for BBB formation (Song et al., 2019). For this study we chose to employ well-characterized and established cell sources to develop stable and standardized assays that provide good cell quality and enable comparisons between experimental groups based on clinically relevant endpoints. We have reedited the Discussion to address this very important point in the revised manuscript (Discussion).

Comment 2. The depth of analysis of alpha-synuclein pathology is very superficial. E.g. Western blot should be used to further characterize pathological a-syn species (p-syn). CC3 stainings are very variable and require additional cell death assays. Astrocyte and microglia activation markers (GFAP, DC68) should be complemented by extensive expression analysis and add a classical neuroinflammatory stimulus for comparison.

Response: Thank you- following the Reviewer's comment we run new experiments and Western blots to characterize the generation of pathological α -syn species (pSer129) in the Brain-Chip.

Added text:

On page 11, line 232-Phosphorylation of α Syn in the brain channel was confirmed by immunoblot analysis at six days upon exposure to α Syn fibrils or PBS (**Fig. 3d**).

On page 15, line 332-The effective phosphorylation of α Syn in the vascular channel was confirmed by immunoblot analysis at six days post exposure to α Syn fibrils as compared to the PBS group (**Supplementary Fig. 6d**).

On page 53, line 1132 Figure 3. d) Western blotting analysis of cell lysates from the Brain channel shows significant intracellular phosphorylation of α Syn at Ser129 (MW: 18kDa) following exposure to α Syn fibrils, whereas there was no effect upon exposure to the PBS. For loading control, equal amounts of protein were immunoblotted with GAPDH antibody (MW: 37kDa).

(Supplementary Information) **On page 11, line 131-Supplementary Figure 6.** d) Western blotting analysis of cell lysates from the vascular channel shows significant intracellular phosphorylation of α Syn at Ser129 (MW: 18kDa) following exposure to α Syn fibrils, whereas there was no effect upon exposure to PBS. For loading control, equal amounts of protein were immunoblotted with GAPDH antibody (MW: 37kDa).

Western blotting. RIPA cell lysis buffer (50 mM Tris, pH 8.0, 150 mM NaCl, 5 mM EDTA, 1% NP-40, 0.5% sodium deoxycholate, and 1% SDS) supplemented with protease and phosphatase inhibitors (Sigma) was used for the extraction of total protein from either brain or vascular channel. The Auto Western Testing Service was provided by RayBiotech, Inc. (Peachtree Corners, GA USA). 0.1 mg/mL sample concentration was loaded into the automated capillary electrophoresis machine. Phosphorylated α Syn at Ser129 was detected with a pSer129- α -syn antibody (23706; Cell Signaling). Gluceraldehyde-3-phosphate dehydrogenase (GAPDH) antibody provided by RayBiotech was used as the loading control. This information has been added in the Methods section of the revised manuscript.

Response: Thank you for the comment. Following the reviewer's suggestion, we have conducted additional experiments to detect apoptosis by TUNEL staining. These new results have been added in Supplementary Information as below.

Added text and images:

On page 12, line 275-Dopaminergic neuron apoptosis was also evaluated and confirmed by TUNEL staining (**Supplementary Fig. 4d, e**).

(Supplementary Information) **On page 7, line 85-Supplementary Figure 4.** d) Representative confocal images showing immunostaining for TH (red) and TUNEL (green) in the brain channel after six days upon-exposure to α Syn fibrils (D8 of the experiment) to α Syn fibrils or α Syn monomers. Scale bars: 100 μ m. e) Percentage of the number of TUNEL positive within the TH-positive neurons. Statistical analysis by

Student's t test (n=3 independent chips/3-4 distinct, randomly selected areas per chip, ****p<0.0001 compared to monomeric group). Error bars represent mean±SEM.

Response: Thank you for your constructive comments. Based on reviewer's suggestion, we complemented the findings on glia activation with gene expression analysis. In particular, we characterized gene expression of specific markers of reactive astrocytes (GFAP, Vimentin, and LCN2) and activated microglia (CD68 and EDA), upregulated in pathological conditions (Kullberg et al., 2001; Walker and Lue, 2015; Li et al, 2019). We found significant upregulation of all the assessed markers at day 8 following exposure to α Syn fibrils compared to the monomeric-treated group (**Supplementary Fig. 5a**). Also, using the same experimental conditions, we treated a separate chips group with TNF- α , a potent neuroinflammatory factor, as a positive control for comparison purposes. In a preliminary study, exposure of the brain channel to TNF- α (100ng/mL) for 72 hrs had an effect similar to that of the α Syn fibrils in gene expression. These new results have been added in the Supplementary Information as below.

Although adding a classical neuroinflammatory stimulus (TNF- α ; 100ng/mL) validate our results, we would like to kindly request the TNF- α treated group not to be included in the final figure since it is a preliminary result of an ongoing study.

Data not shown: Gene expression of reactive astrocyte (GFAP, Vimentin, and LCN2) and activated microglia markers (CD68 and EDA) in the brain channel six days post α Syn fibrils or 72 hr post TNF- α exposure. Statistical analysis by Student's t-test (n=3 independent chips, *p < 0.05 and **p < 0.01 compared to TNF- α). Error bars represent mean±SEM.

Added text and images:

On page 13, line 278-We performed quantitative analysis on the gene expression of markers for reactive astrocyte and activated microglia, that are upregulated in pathological conditions. In particular, we found that six days post exposure of the Substantia Nigra Brain-Chip to α Syn fibrils, both reactive astrocyte markers (GFAP, Vimentin, and LCN2) and activated microglia markers (CD68 and EDA) were upregulated at compared to the monomeric α Syn-treated group (**Supplementary Fig. 5a**).

(Supplementary Information) **On page 8, line 95-Supplementary Figure 5.** a) Gene expression of specific markers for reactive astrocyte (GFAP, Vimentin, and LCN2) and activated microglia (CD68 and EDA) in the brain channel at day six post exposure to α Syn fibrils or α Syn monomers. Statistical analysis by Student's t-test ($n=3$ independent chips, $*p < 0.05$ and $**p < 0.01$ compared to α Syn monomeric group). Error bars present $\text{mean} \pm \text{SEM}$.

RNA isolation, reverse transcription, and qPCR. Each channel of the Brain-Chip was lysed using TRI Reagent® (Sigma- Aldrich, T9424) and RNA was isolated using the Direct-zol RNA Purification Kit (Zymo Research, R2060). Genomic DNA was removed using the TURBO DNA-free™ Kit (ThermoFischer, AM1907) and reverse transcription to cDNA was performed with the SuperScript IV Synthesis System (ThermoFischer, 18091050). TaqMan Fast Advanced Master Mix (Applied Biosystems, 4444963) and TaqMan Gene Expression Assays were used for the Quantitative Real Time PCR (qPCR). The qPCR was detected on a QuantStudio 3 PCR System (Fisher Scientific). The target genes were assessed using commercially available primers (GFAP: Hs00909233_m1, VIM: Hs00958111_m1, LCN2: Hs01008571_m1, CD68: Hs02836816_g1, EDA: Hs03025596_s1, APOA1: Hs00163641_m1, SNCAIP: Hs00917422_m1, LRRK2: Hs01115057_m1, MAOA: Hs00165140_m1). All primers are also listed in **Supplementary Table 2**. The results were quantified by the comparative Ct method. Ct values for samples were normalized to the expression of the housekeeping gene (18s: Hs99999901_s1; Applied Biosystems).

Reviewer #3 (Remarks to the Author)

This manuscript leveraged Organs-on Chips technology to engineer a human Brain Chip representative of the substantia nigra area of the brain, which is most affected in Parkinson's disease. The α -syn fibril model was capable of reproducing key aspects of Parkinson's including phosphorylated α -syn, mitochondrial impairment, neuroinflammation and compromised barrier function. Therefore is proposed to serve as a platform for novel therapeutic interventions.

This manuscript builds on the already published work from this group where they developed a human disease model of drug toxicity-induced pulmonary edema in a lung on a chip microdevice.

This is a very interesting model system which could be potentially very useful to the Parkinson's community and the field as a whole if the issues below are addressed:

Further information on some of the cells used should be provided: the authors mention primary human astrocytes and microglia, there is no information on the age of the donor or the brain region from which the microglia came from. Were these collected from elective abortions? Were these cells expanded in culture prior to being used in experiments? This information would be useful in determining how akin this set up should be to the adult midbrain which is affected in Parkinson's.

The authors should confirm midbrain dopamine neuron identity in Figure 1. They state on page 5, line 106, that 'First (D0) we seeded in the brain channel human iPS-derived DA neurons' so how many days post neutralisation were these cells? I note in the reporting summary that midbrain markers were used but its not clear if this is by the company who sold them or by the authors themselves.

In the methods there are medias defined as DeSR2 and DeSR1, what are these exactly?

Figure 5a, CC3 staining would appear to be more widespread than the level detailed in the graph on the right, authors should consider using separate channel images to show this more clearly.

On page 21, line 459 there is a reference to Chips were functionalised using Emulate's proprietary reagents. The whole idea of publishing is so that other researchers can benefit by increased scientific knowledge within the scientific community. However, nobody could try to replicate these results without the knowledge of such reagents? This is really not within the spirit of publishing scientific research.

Page 21, line 471 BMEC were seeded at 16-20 million cells per ml. This is a lot of cells and authors should comment on whether this is a rate limiting factor for using this technology?

Responses to Reviewer #3

We thank the reviewer for his/her constructive comments, which have led to a substantial improvement in our manuscript. In particular, we have:

- 1) Performed further analysis to characterize the closeness of the transcriptomic signatures between the Brain-Chip and adult human midbrain.
- 2) Confirmed the midbrain dopamine neuron identity of the iPSC-derived neurons used in our studies.
- 3) Evaluated the vulnerability of dopaminergic neurons to α Syn fibril-induced Brain-Chip model.

Please find below detailed responses to each specific comment.

Comment 1. Further information on some of the cells used should be provided: the authors mention primary human astrocytes and microglia, there is no information on the age of the donor or the brain region from which the microglia came from. Were these collected from elective abortions? Were these cells expanded in culture prior to being used in experiments? This information would be useful in determining how akin this set up should be to the adult midbrain which is affected in Parkinson's.

Response: Thank you for your questions that gave us the opportunity to provide details on this important subject. In all our described studies we have used only company-validated, commercially available cell lines. Specifically, we employed primary human astrocytes and microglia validated by a number of prior studies in the field. Consistently with the existing literature and the vendors' Certificate of Analysis, we confirmed by immunostaining the expression of important markers, such as GFAP (glial fibrillary acidic protein) for mature astrocytes, or IBA1 and TMEM119 for mature microglia. In reference to the human primary microglia, the HMC3 cell line used was established through SV40-driven immortalization of a human fetal cortex-derived primary microglia culture. As described in the original paper from Marc Tardieu's lab, these cells have originated from material obtained from tissue derived from elective abortions of eight-to ten-week-old embryos, in compliance with the recommendations of the French National Ethics Committee and approval by the local ethics committee (Janabi et al. 1995). For our study, human primary cells were sub-cultured twice in order to obtain a homogenous population for experimental use and stored in frozen aliquots at passage 3 (p3).

To provide further evidence that the Brain-Chip could capture the expression pattern of the adult substantia nigra better from the conventional cell culture system, we identified a list of 13 genes, characteristic of the specific cell types used to build our model (**Fig. 2e**). For each one of these 13 genes, we compared fold change in expression levels in the following groups: i) Adult Substantia Nigra vs. Brain-Chip and ii) Adult Substantia Nigra vs. Conventional Cell Culture system. Our results show that the expression levels of all these genes in the Brain-Chip are more similar to those in the adult substantia nigra, as compared to the conventional cell culture systems (smaller $|\text{Log}_2(\text{Fold Change})|$ values). Notably, for most of these genes (as indicated with asterisks in **Fig. 2e**) the

expression changes in the Brain-Chip were significantly smaller as compared to those of the conventional cell culture system.

These results indicate that the gene expression patterns that characterize the primary cell types of the substantia nigra brain tissue are better recapitulated in our Brain-Chip model than in conventional culture systems. In the revised manuscript, we have added a paragraph to discuss all these new findings.

Added text:

On page 9, line 203-To provide further evidence that the Brain-Chip could capture the expression pattern of the adult substantia nigra better from the conventional cell culture system, we identified a list of 13 genes, characteristic of the specific cell types used to build our model (**Fig. 2e**). For each one of these 13 genes, we compared fold change in expression levels in the following groups: i) Adult Substantia Nigra vs. Brain-Chip and ii) Adult Substantia Nigra vs. Conventional Cell Culture system. Our results show that the expression levels of all these genes in the Brain-Chip are more similar to those in the adult substantia nigra, as compared to the conventional cell culture systems (smaller $|\text{Log}_2(\text{Fold Change})|$ values). Notably, for most of these genes (as indicated with asterisks in **Fig. 2e**) the expression changes in the Brain-Chip were significantly smaller as compared to those of the conventional cell culture system.

On page 51, line 1113-Figure 2. e) Gene expression pattern analysis between the following groups: i) Adult Substantia Nigra vs. Brain-Chip and ii) Adult Substantia Nigra vs. Conventional Cell Culture system. In the Brain-Chip, most of these genes (those with an asterisk) exhibit statistically significantly smaller fold changes ($p < 0.01$) in their expression levels as compared to the conventional cell culture system.

Comment 2. The authors should confirm midbrain dopamine neuron identity in Figure 1. They state on page 5, line 106, that ‘First (D0) we seeded in the brain channel human iPSC-derived DA neurons’ so how many days post neutralisation were these cells? I note in the reporting summary that midbrain markers were used but its not clear if this is by the company who sold them or by the authors themselves.

Response: According to the selling company, human iPSC-derived dopaminergic neurons (iCell® Neurons) is a highly pure, mature, population of fully differentiated neurons with a midbrain dopaminergic specificity. They are ready-to-use neurons (D0), that develop branched networks and exhibit high levels of neuronal activity, as soon as day 2 post-plating (D2). Following the reviewer’s suggestion, we have conducted experiments to confirm the expression of additional midbrain dopaminergic neuron markers such as LMX1A, FOXA2, and TH, at D2 and D8 of culture. Specifically, >80% of the TH positive neurons in the Brain-Chips were FOXA2 and LMX1A positive, indicative of their floorplate origin. These additional images have been added in the Supplementary Information (**Supplementary Fig. 2e, f**). As per our response to **Comment 1** above, gene expression analysis confirmed the midbrain dopaminergic neuron identity of the cells used. These new results have been added in the Supplementary Information of the revised manuscript, as below.

Added text and images:

On page 6, line 131-Furthermore, the majority of the TH-positive neurons were positive for both FOXA2 (day 2: 89.5 ± 1.7 and day 8: $92.8 \pm 1.6\%$), and LMX1A (day 2: 88.3 ± 0.8 and day 8: $90 \pm 1.5\%$) over the course of the culture, indicative of their midbrain floorplate origin (**Supplementary Fig.2 e, f**).

(Supplementary Information) **On page 4, line 50-Supplementary Figure 2.** e) Double-label immunofluorescence with antibodies against TH, FOXA2, and LMX1A at day 2 and 8. f) Percentage of FOXA2 and LMX1A positive neurons within the MAP2 and TH-positive population, in line with their floorplate midbrain phenotype (n=3 Brain-Chips with 3~5

randomly selected different fields of view per chip, NS: Not Significant). Error bars represent mean±SEM. Scale bars: 100 μm.

Comment 3. In the methods there are medias defined as DeSR2 and DeSR1, what are these exactly?

Response: We have used a published protocol to directly differentiate human pluripotent stem cells to blood-brain barrier endothelial cells (Qian et al., 2017). The DeSR1 medium is composed of DMEM/Ham's F12 (Thermo Fisher Scientific), 1× MEM-NEAA (Thermo Fisher Scientific), 0.5× GlutaMAX (Thermo Fisher Scientific), and 0.1 mM β-mercaptoethanol (Sigma). After 24 hours, the medium was replaced by DeSR2 medium that is composed by DeSR1 plus 1× B27 (Thermo Fisher Scientific); the medium was refreshed every day for a period of another five days. This information has been added to the Methods.

Comment 4. Figure 5a, CC3 staining would appear to be more widespread than the level detailed in the graph on the right, authors should consider using separate channel images to show this more clearly.

Response: We thank the reviewer for this comment. We separated the channel images to clearly show the CC3 staining. The selective vulnerability of dopaminergic neurons in our αSyn fibrils-induced model, was also assessed by caspase-3 activation in the TH-positive neurons. These new results have been added in the revised manuscript in the Results section, as below.

Added text and Images:

On page 12, line 271-Confocal immunocytochemical analysis using antibodies against microtubule-associated protein 2 (MAP2), tyrosine hydroxylase (TH), and cleaved caspase-3 (CC3), confirmed the increase in caspase 3-positive dopaminergic neurons at day six post exposure to αSyn fibrils, as compared to those exposed to monomeric αSyn (Fig. 5a, b).

On page 56, line 1182-Figure 5. a) Representative merged images showing immunostaining for MAP2 (grey), tyrosine hydroxylase (red, TH), and Cleaved Caspase-3 (green, CC3) in the brain channel at six-days post-exposure to αSyn fibrils or αSyn

monomers. Scale bars: 100 μm . b) Quantitation of the number of CC3 and MAP2- or TH-positive neurons. Statistical analysis by two-way ANOVA with Tukey's multiple comparisons test (3~4 randomly selected different areas per chip, n=3 independent chips/experimental group, **p<0.01, ****p<0.0001, NS: Not Significant). Error bars represent mean \pm SEM. Scale bars: 100 μm .

Comment 5. On page 21, line 459 there is a reference to Chips were functionalised using Emulate's proprietary reagents. The whole idea of publishing is so that other researchers can benefit by increased scientific knowledge within the scientific community. However, nobody could try to replicate these results without the knowledge of such reagents? This is really not within the spirit of publishing scientific research.

Response: Thank you for pointing this out and gives us the opportunity to add some information on Emulate's proprietary reagents used, i.e., ER-1 that refers to a surface activation reagent and ER-2 to a resuspension and wash buffer. Both reagents are commercially available by Emulate Inc. Briefly, ER-1 (Emulate reagent: 10461) and ER-2 (Emulate reagent: 10462) are mixed at a concentration of 1mg/ml before being added to the top and bottom microfluidic channels of the chip. The platform is then irradiated with high power UV light having peak wavelength of 365nm and intensity of 100 $\mu\text{J}/\text{cm}^2$ for 20min using a UV oven (CL-1000 Ultraviolet Crosslinker AnalytiK-Jena: 95-0228-01). This information has been added in the revised manuscript in the Methods section.

Comment 6. Page 21, line 471 BMEC were seeded at 16-20 million cells per ml. This is a lot of cells and authors should comment on whether this is a rate limiting factor for using this technology?

Response: The endothelial cells' density required is not a limiting factor since iPSC-derived BMECs are expanded post differentiation in flasks for two days, and the obtained cell numbers are enough for the seeding of 48 Brain-Chips. In our present study, the seeded density of BMECs is lower or similar to other BBB models being used (4-20 million cells per ml (Vatine et al., 2019; Jagadeesan et al., 2020), 23 million cells per ml (Park et al., 2019).

References

- Ahn, S. I. et al. Microengineered human blood–brain barrier platform for understanding nanoparticle transport mechanisms. *Nat. Commun.* 11, (2020).
- Chrobak, I., Lenna, S., Stawski, L. & Trojanowska, M. Interferon- γ promotes vascular remodeling in human microvascular endothelial cells by upregulating endothelin (ET)-1 and transforming growth factor (TGF) β 2. *J. Cell. Physiol.* 228, 1774–1783 (2013).
- Chung, H. K., Ho, H.-A., Pérez-Acuña, D. & Lee, S.-J. Modeling α -Synuclein Propagation with Preformed Fibril Injections. *J. Mov. Disord.* 12, 139–151 (2019).
- Courte, J. et al. The expression level of alpha-synuclein in different neuronal populations is the primary determinant of its prion-like seeding. *Sci. Rep.* 10, (2020).
- De Miranda, B. R., Rocha, E. M., Castro, S. L. & Greenamyre, J. T. Protection from α -Synuclein induced dopaminergic neurodegeneration by overexpression of the mitochondrial import receptor TOM20. *npj Park. Dis.* 6, (2020).
- Debnath, K., Pradhan, N., Singh, B. K., Jana, N. R. & Jana, N. R. Poly(trehalose) Nanoparticles Prevent Amyloid Aggregation and Suppress Polyglutamine Aggregation in a Huntington's Disease Model Mouse. *ACS Appl. Mater. Interfaces* 9, 24126–24139 (2017).
- Echigo, R. et al. Trehalose treatment suppresses inflammation, oxidative stress, and vasospasm induced by experimental subarachnoid hemorrhage. *J. Transl. Med.* 10, (2012).
- Galet, B., Cheval, H. & Ravassard, P. Patient-Derived Midbrain Organoids to Explore the Molecular Basis of Parkinson's Disease. *Frontiers in Neurology* vol. 11 (2020).
- Hoffmann, A.-C. et al. Extracellular aggregated alpha synuclein primarily triggers lysosomal dysfunction in neural cells prevented by trehalose. *Sci. Rep.* 9, 544 (2019).
- Imamura, K. et al. Distribution of major histocompatibility complex class II-positive microglia and cytokine profile of Parkinson's disease brains. *Acta Neuropathol.* 106, 518–526 (2003).
- Jagadeesan, S., Workman, M. J., Herland, A., Svendsen, C. N. & Vatine, G. D. Generation of a human iPSC-based blood-brain barrier chip. *J. Vis. Exp.* 2020, (2020).
- Janabi, N., Peudenier, S., Héron, B., Ng, K. H. & Tardieu, M. Establishment of human microglial cell lines after transfection of primary cultures of embryonic microglial cells with the SV40 large T antigen. *Neurosci. Lett.* 195, 105–108 (1995).
- Kasendra, M. et al. Duodenum Intestine-Chip for preclinical drug assessment in a human relevant model. *Elife* 9, (2020).
- Khalifeh, M., Barreto, G. E. & Sahebkar, A. Trehalose as a promising therapeutic candidate for the treatment of Parkinson's disease. *British Journal of Pharmacology* vol. 176 1173–1189 (2019).
- Khalifeh, M., Read, M. I., Barreto, G. E. & Sahebkar, A. Trehalose against Alzheimer's Disease: Insights into a Potential Therapy. *BioEssays* 42, (2020).
- Klein, C. & Westenberger, A. Genetics of Parkinson's disease. *Cold Spring Harb. Perspect. Med.* 2, (2012).
- Kullberg, S., Aldskogius, H. & Ulfhake, B. Microglial activation, emergence of ED1-expressing cells and clusterin upregulation in the aging rat CNS, with special reference to the spinal cord. *Brain Res.* 899, 169–186 (2001).
- Larocca, T. J. et al. Translational evidence that impaired autophagy contributes to arterial ageing. *J. Physiol.* 590, 3305–3316 (2012).

Li, K., Li, J., Zheng, J. & Qin, S. Reactive astrocytes in neurodegenerative diseases. *Aging and Disease* vol. 10 664–675 (2019).

Marchetti, L. & Engelhardt, B. Immune cell trafficking across the blood-brain barrier in the absence and presence of neuroinflammation. *Vasc. Biol.* 2, H1–H18 (2020).

Mount, M. P. et al. Involvement of interferon- γ in microglial-mediated loss of dopaminergic neurons. *J. Neurosci.* 27, 3328–3337 (2007).

Ng, C. T. et al. Interferon-Gamma Increases Endothelial Permeability by Causing Activation of p38 MAP Kinase and Actin Cytoskeleton Alteration. *J. Interf. Cytokine Res.* 35, 513–522 (2015).

Pagano, K., Tomaselli, S., Molinari, H. & Ragona, L. Natural Compounds as Inhibitors of A β Peptide Aggregation: Chemical Requirements and Molecular Mechanisms. *Frontiers in Neuroscience* vol. 14 (2020).

Park, T. E. et al. Hypoxia-enhanced Blood-Brain Barrier Chip recapitulates human barrier function and shuttling of drugs and antibodies. *Nat. Commun.* 10, (2019).

Patel, R., Page, S. & Al-Ahmad, A. J. Isogenic blood–brain barrier models based on patient-derived stem cells display inter-individual differences in cell maturation and functionality. *J. Neurochem.* 142, 74–88 (2017).

Qian, T. et al. Directed differentiation of human pluripotent stem cells to blood-brain barrier endothelial cells. *Sci. Adv.* 3, e1701679 (2017).

Rivetti di Val Cervo, P., Besusso, D., Conforti, P. & Cattaneo, E. hiPSCs for predictive modelling of neurodegenerative diseases: dreaming the possible. *Nat. Rev. Neurol.* 1–12 (2021) doi:10.1038/s41582-021-00465-0

Serrano-Pozo, A. et al. Differential relationships of reactive astrocytes and microglia to fibrillar amyloid deposits in alzheimer disease. *J. Neuropathol. Exp. Neurol.* 72, 462–471 (2013).

Song, L. et al. Functionalization of Brain Region-specific Spheroids with Isogenic Microglia-like Cells. *Sci. Rep.* 9, (2019).

Vatine, G. D. et al. Human iPSC-Derived Blood-Brain Barrier Chips Enable Disease Modeling and Personalized Medicine Applications. *Cell Stem Cell* 24, 995-1005.e6 (2019).

Walker, D. G. & Lue, L. F. Immune phenotypes of microglia in human neurodegenerative disease: Challenges to detecting microglial polarization in human brains. *Alzheimer's Research and Therapy* vol. 7 (2015).

REVIEWER COMMENTS

Reviewer #1 (Remarks to the Author):

The authors have satisfactorily addressed most of my prior comments.

Reviewer #2 (Remarks to the Author):

In general, the authors addressed several remarks brought up by the reviewers by providing additional data. However, some major concerns still arise that have not been addressed sufficiently:

One major concern is the reproducibility of this study with other cell lines. While I understand the author's preference for well characterized cell lines and therefore the lack of isogenic pairs, the data analysis runs short, when only this one cell line is analyzed for RNAseq. Therefore, in my view additional pairs of lines should be carried out to render this N=1 study meaningful. Even if the bioinformatics bulk RNA analyses of the three groups was carried out, technical and biological artifacts of the RNA-Seq measurements cannot be extracted from one cell line only.

Moreover, questions arise regarding the claim that brain-chip technology more closely resembles human substantia nigra based on the bulk RNA-seq data provided. While there is an overlap of DE genes both brain-chip as well as adult substantia nigra when compared to a conventional cell culture approach (Fig. 2 d), this overlap represents only a minority of the DE genes. Furthermore, in Fig. 2e, the authors then switch towards the fold expression levels of different key marker genes for different cell types out of the bulk RNA-seq analysis, thereby only providing data for a small gene subset. In their response to Reviewer 1, the authors report that on a global transcriptomic scale, the direct comparison between the transcriptome of i) Adult Substantia Nigra vs. Brain-chip as well as ii) Adult Substantia Nigra vs. Conventional Cell Culture yields 566 and 682 DE genes, respectively. While the total number of DE genes seems to be lower in analysis i), this data in my opinion also points towards substantial differences of both a conventional culture system as well as the brain chip technology compared to adult SN on a bulk RNA-seq level. I think the authors should discuss this circumstance more critically. For further clarity, I would recommend the authors to include an illustration depicting the number of significantly DE genes of comparisons i and ii, respectively, either in the main figure or the supplements. Adding additional lines to the experiment may also enable the authors to perform PCA analyses on a global transcriptomic scale between different adult SN specimens as well as different conventional culture systems and brain-chips.

Additionally, questions arise regarding the specific clinical relevance of the experiments performed by incubating the brain-chip assembly with synuclein fibrils.

On the one hand, the synuclein preparation the authors claimed to use may contain extensive residual amounts of Endotoxin as the manufacturer only guarantees a level of <20000 EU/ml, an amount still substantial enough to trigger an LPS mediated inflammatory response especially in the microglia-like cells in the co-culture system. While the authors stated in the methods part that they controlled for LPS contamination by measurement using the LAL assay, this data should be depicted in a supplemental figure or material in order better interpret confounding immunogenic properties of the used preparations.

Moreover, additional minor concerns are:

- In Fig. 1 f, the authors present the permeability to 3kDA fluorescent dextran and 0.5kDA lucifer yellow as a quantification of barrier function at different time points. However, a statistical analysis comparing the different time points is not depicted in the graph. Is the permeability at day 8 significantly lower compared to day 5?

Reviewer #3 (Remarks to the Author):

The authors have provided a substantial amount of new data and in so doing have significantly improved the manuscript. They have answered my queries and I now recommend publication.

Responses to Reviewers

Reviewer #1

General Comments:

The authors have satisfactorily addressed most of my prior comments.

Response: We thank for the reviewer for his/her positive recommendation.

Reviewer #2

In general, the authors addressed several remarks brought up by the reviewers by providing additional data. However, some major concerns still arise that have not been addressed sufficiently:

We thank the reviewer for his/her constructive comments, which have led to a substantial improvement in our manuscript. Please see below our point-by-point response (underlined text below indicates the reviewer's comments) together with new supportive data from additional, recently performed experiments:

Comment 1: One major concern is the reproducibility of this study with other cell lines. While I understand the author's preference for well characterized cell lines and therefore the lack of isogenic pairs, the data analysis runs short, when only this one cell line is analyzed for RNAseq. Therefore, in my view additional pairs of lines should be carried out to render this N=1 study meaningful. Even if the bioinformatics bulk RNA analyses of the three groups was carried out, technical and biological artifacts of the RNA-Seq measurements cannot be extracted from one cell line only.

Response: We thank the reviewer for the comment. The quality and sensitivity of RNA-sequencing technology have improved greatly over the past decade, making it a highly reliable method for characterization of genome-wide transcriptome (also noted by SEQC Consortium, 2016). In the present study, we used four biological samples per group eliminating any technical and biological artifacts of the RNA-Seq measurements. As suggested by the reviewer, we assessed the robustness and reproducibility of our model using a different set of lines (**Table below**). Next, we compared the global RNA-sequencing (RNA-seq) data of neurovascular units constructed in the Brain-Chips cultured under constant flow (n=4) and in conventional cell cultures (n=4), the most widely used static in vitro system. Both systems were seeded using the same cell-type composition and subjected to the same experimental conditions (see Material and Methods).

We first performed differential gene expression (DGE) analysis between the Brain-Chip and the conventional cell cultures. We applied the following thresholds to select the differentially expressed (DE) genes: adjusted p-value<0.05 and $|\log_2\text{FoldChange}| > 1$. Out of the 38,887 genes annotated in the genome, 1035 were significantly differentially expressed in the Brain-Chips, with 610 and 425 genes up- or down-regulated respectively. These findings are consistent with the original set of data obtained with different cell lines, where 1316 were significantly differentially expressed in the Brain-Chips, with 646 and 670 genes up- or down-regulated, respectively. Then, we performed Gene Ontology analysis utilizing the Gene Ontology knowledgebase, to highlight the biological processes significantly enriched in these gene sets. Among the up-regulated genes in the Brain-Chip samples, we identified functional gene sets that significantly clustered under 75 GO terms. These functional gene sets were part of brain-relevant biological processes, including synaptic pruning, transmembrane transport, extracellular matrix organization, cell adhesion, tissue development, and response to stimulus. Compared to the Brain-Chip, the transcriptome of the conventional cell cultures was enriched in genes involved in anterior/posterior pattern specification, embryonic and

tissue morphogenesis, cell differentiation, negative regulation of cellular processes. These findings suggest that the cells cultured in the Brain-Chip may obtain a more mature and/or differentiated state compared to the cells maintained in the conventional cell cultures. These results are in line with the data obtained from the 1st set of donor cells (included in the original submission), showing that cells in the Brain-Chip reach higher resemblance to the biological characteristics of the mature tissue as compared to conventional cell cultures.

Aside from the RNA analysis, we also repeated the α Syn fibrils-induced challenge in Brain-Chips seeded with a different, 2nd set, of cell lines to further support our original findings. As shown below we were able to reproduce the key functional endpoints of our model such as phosphorylated α Syn (pSer129- α Syn), glial activation (neuroinflammation, tight junction disruption, and the compromised blood brain barrier).

As these data confirm the findings and associated conclusions provided in the original submission, to avoid redundancy they have not been added in the revised manuscript; instead, we have included a sentence (Results) to highlight that the shown results have been reproduced with an additional, distinct set of cell lines.

Added text:

On page 8, line 183 -Notably, there were no significant variations observed across Brain-Chips reproduced with an additional, distinct set of cell lines (data not shown).

Brain-Chip (2nd set of lines)	Vendor	Catalog (Lot) Number
Human iPSC-derived DA neurons	iXCells	40HU-012 (400348-DAN-M59C)
Human Primary Astrocytes	NeuCyte	1010 (13029-017)
Human iPSC-derived Microglia	Cellular Dynamics	C1110 (104683)
Human Primary Pericytes	Cell Systems	ACBRI 498 (498.03.02.01 10)
Human iPSCs (Brain Endothelial cells)	iXCells	400255 (400255-2)

Table (2nd set of cell lines): The conventional cell cultures and the Brain-Chips were seeded using the same ECM composition as well as cell elements and seeding density as described in the Material and Methods for the 1st set of lines. For the apical compartment we used complete maintenance media (iXCells Biotechnologies), while for the basolateral compartment we used HBMEC medium.

Brain-Chip (2nd set of lines)

a

b

c

Differentially Expressed (DE) genes and enriched gene ontology categories in Brain-Chips as compared to conventional cell cultures (2nd set of cell lines).

a) The volcano plot resulted by the DGE analysis between Brain-Chip and conventional cell cultures using the 2nd set of lines. For the selection of the DE genes, we used the following thresholds: adjusted p-value < 0.05 and |Log₂(foldchange)| > 1. The identified up- (down-) regulated genes are highlighted in cyan (magenta) color respectively. Sample sizes were as follows: Brain-Chip, n=4, conventional cell culture system, n=4. b) and c) List of biological processes identified by Gene Ontology (GO) enrichment analysis using the up- and down-regulated genes respectively resulted by the differentially gene expression analysis between Brain-Chip and conventional cell cultures. Culture in Brain-Chips and conventional cell cultures were done in parallel. Samples were collected and processed for analyses 8 days post-seeding (D8).

Brain-Chip (2nd set of lines)

Key pathological features in α Syn fibrils-induced Brain-Chip model (2nd set of cell lines). (a-c) Immunofluorescence micrographs show accumulation of phosphorylated α Syn (green, phospho- α Syn129; red, ZO-1; blue, DAPI) in the brain and vascular channel of the Brain-Chip, following treatment with α Syn fibrils, as opposed to α Syn monomeric group (scale bars: 100 μ m). Quantitative analysis of fluorescence intensity in each group at day six post exposure to α Syn fibrils or α Syn monomers. Statistical analysis by Student's t-test ($n=4$ independent chips with 5~6 randomly selected different areas per chip, **** $P<0.0001$ compared to monomeric group). Error bars represent mean \pm SEM. d) (Upper micrographs) Immunostaining for the astrocyte marker GFAP (magenta) demonstrated activation of astrocytes at day six post exposure to α Syn fibrils compared to monomeric α Syn (scale bar, 100 μ m). (Lower micrographs) Immunostaining for the microglial CD68 (red) demonstrated activation of astrocytes and microglia at day six post exposure to α Syn fibrils compared to monomeric α Syn (scale bar, 100 μ m). e) Quantitative barrier function analysis as per the apparent permeability to 0.5 kDa Lucifer yellow at day 8 following exposure to α Syn fibrils or α Syn monomers. Statistical analysis by Student's t-test ($n=3\sim 4$ independent chips, ** $P<0.01$ compared to monomeric group).

Comment 2: Moreover, questions arise regarding the claim that brain-chip technology more closely resembles human substantia nigra based on the bulk RNA-seq data provided. While there is an overlap of DE genes both brain-chip as well as adult substantia nigra when compared to a conventional cell culture approach (Fig. 2 d), this overlap represents only a minority of the DE genes. Furthermore, in Fig. 2e, the authors then switch towards the fold expression levels of different key marker genes for different cell types out of the bulk RNA-seq analysis, thereby only providing data for a small gene subset. In their response to Reviewer 1, the authors report that on a global transcriptomic scale, the direct comparison between the transcriptome of i) Adult Substantia Nigra vs. Brain-chip as well as ii) Adult Substantia Nigra vs. Conventional Cell Culture yields 566 and 682 DE genes, respectively. While the total number of DE genes seems to be lower in analysis i), this data in my opinion also points towards substantial differences of both a conventional culture system as well as the brain chip technology compared to adult SN on a bulk RNA-seq level. I think the authors should discuss this circumstance more critically. For further clarity, I would recommend the authors to include an illustration depicting the number of significantly DE genes of comparisons i and ii, respectively, either in the main figure or the supplements. Adding additional lines to the experiment may also enable the authors to perform PCA analyses on a global transcriptomic scale between different adult SN specimens as well as different conventional culture systems and brain-chips.

Response: Thank you for the comment. We agree with the reviewer's comment that our data points towards substantial transcriptomic differences between the conventional culture system or the Brain-Chip technology, as compared to adult SN. This observation is supported by the PCA analysis and the absolute number of DE genes between different adult SN specimens, conventional culture systems and Brain-Chips (**Supplementary Figure 1d,e**).

In the present study, our primary focus was to highlight the important transcriptomic differences between Brain-Chip and conventional cell culture systems, important attributes of our new platform developed for the study of human brain diseases. Importantly, as suggested by the reviewer, we have run new experiments that demonstrated the consistency of our findings among Brain-Chips derived from different sets of cells in support of the system's reliability (**Comment 1**).

The adult substantia nigra tissue was used as a reference standard (known transcriptomic reference) for comparison purposes between the two experimental technologies (Organ-Chip vs conventional culture system). Although we identified slight differences between the number of DE genes (absolute numbers) in conventional culture system and Brain-Chip compared to adult substantia nigra (**Supplementary Figure 1b**), the expression levels of characteristic cell signature specific genes in the adult substantia nigra were closer to those in the Brain-Chip rather than to those in the conventional cell culture systems (**Figure 2e**). An example is demonstrated through comparison of the expression levels of 13 genes related to the cell maturation state, specifically picked to highlight the different state of cells that are important for disease modeling when cultured in the Brain-Chip vs in conventional cell culture systems. Our results also support data from several other groups including ours on the contribution of the microphysiological, microfluidic

systems platforms towards the development of well-established in vitro systems aiming for better modeling of human brain and the pathogenesis of neurodegeneration.

In agreement with the reviewer's suggestion, we have included an illustration depicting the number of significantly DE genes of comparisons i and ii, respectively (**Supplementary Figure 1d,e**).

Added text and images:

On page 9, line 205- We next performed a direct comparison of the RNA sequencing profiles by comparing DGE analyses between the following groups: i) Adult Substantia Nigra vs. Brain-Chip and ii) Adult Substantia Nigra vs. Conventional Cell Culture system. The first analysis revealed 566 significantly DE genes in adult substantia nigra, 186 up- and 380 down-regulated. The second analysis revealed 682 DE genes in adult substantia nigra 205 up- and 477 down-regulated (**Supplementary Figure 1d**). In line with these findings, principal components analysis (PCA) shows clear separation between the samples of the: (i) conventional culture systems, (ii) Brain-Chip technology and (iii) adult SN (**Supplementary Figure 1e**) which indicates the transcriptomic differences between these groups. Although the numbers of the DE genes (absolute number) between the conventional culture system or the Brain-Chip as compared to adult substantia nigra were similar (**Supplementary Figure 1d**), the expression levels of a set of 13 cell signature-specific genes characteristic of the mature tissue in the adult substantia nigra were much closer to those in the Brain-Chip rather than to those in the conventional cell culture systems (**Figure 2e**). For each of these 13 genes, we compared the fold changes in expression levels (smaller $|\text{Log}_2(\text{Fold Change})|$ values) in the following groups: i) Adult Substantia Nigra vs. Brain-Chip and ii) Adult Substantia Nigra vs. Conventional Cell Culture system (**Fig. 2e**).

(Supplementary Information) **On page 2, line 14-Supplementary Figure 1.** d) Comparison of significantly up- or down-regulated genes between the following groups: i) Adult Substantia Nigra vs. Brain-Chip and ii) Adult Substantia Nigra vs. Conventional Cell Culture system (adjusted p -value < 0.05 and $|\log_2\text{FoldChange}| > 1$). e) PCA generated using the RNA-seq data of cells from the conventional culture system (grey) ($n=4$), the Brain-Chip (blue) ($n=4$), and Adult SN (magenta) ($n=8$). A 2D-principal component plot is shown with the first component along the X-axis and the second along the Y-axis. The proportion of explained variance is shown for each component.

These results were consistent with our findings obtained from experiments in Brain-Chips seeded with a different, 2nd set of cell lines:

The principal components analysis (PCA) shows clear separation between the samples of the: (i) conventional culture systems, (ii) Brain-Chip technology and (iii) adult SN (**Supplementary Figure 1e**) which indicates the transcriptomic differences between these groups. However, the expression levels of cell-specific signature genes, characteristic of mature tissue (Neurons; TH and LMX1A, Astrocytes; GFAP and GLAST, Microglia; TREM2 and TMEM119), in the Brain-Chip were more like those in the adult substantia nigra, as compared to the conventional cell culture systems.

a) PCA generated using the RNA-seq data generated by samples from the conventional culture system (grey) (n=4), the Brain-Chip (blue) (n=4), and Adult SN (magenta) (n=8). A 2D-principal component plot is shown with the first component along the X-axis and the second along the Y-axis. The proportion of explained variance is shown for each component. b) Gene expression pattern analysis between the following groups: i) Adult Substantia Nigra vs. Brain-Chip and ii) Adult Substantia Nigra vs. Conventional Cell Culture system. Asterisks indicate statistically significant differences in fold changes (adjusted $P_{val} < 0.05$) in the gene expression levels between the Brain-Chip and the conventional cell culture systems.

As these data confirm the findings and associated conclusions provided in the original submission, to avoid redundancy they have not been added in the revised manuscript; instead, we have included a sentence (Results) to highlight that the shown results have been reproduced with an additional, distinct set of cell lines.

Comment 3: Additionally, questions arise regarding the specific clinical relevance of the experiments performed by incubating the brain-chip assembly with synuclein fibrils. On the one hand, the synuclein preparation the authors claimed to use may contain extensive residual amounts of Endotoxin as the manufacturer only guarantees a level of <20000 EU/ml, an amount still substantial enough to trigger an LPS mediated inflammatory response especially in the microglia-like cells in the co-culture system. While the authors stated in the methods part that they controlled for LPS contamination by measurement using the LAL assay, this data should be depicted in a supplemental figure or material in order better interpret confounding immunogenic properties of the used preparations.

Response: We agree with the reviewer. According to the providing company (abcam), recombinant α -synuclein (either in the monomer or fibril form) contains an average of 10-20 EU/mL (<20.000 EU/mL) of endotoxin. Endotoxin levels were evaluated by the Bacterial Endotoxin Test and Lonza's Quality Systems requirements (Endotoxin Testing Services, Lonza Europe). According to the Endotoxin Testing Report provided by Lonza, our samples were tested at dilutions, 1:10, 1:100, and 1:1000. As positive product control (PPC) recovery in all dilutions applied was valid (PPC must be 50-200%), we used the values in the least manipulated dilution/sample, i.e., the 1:10 dilution, and showed that in all samples measured (α Syn monomers and α Syn fibrils) endotoxin concentration was <0.05 EU/mL.

Sample	Dilutions	Positive Product Control (PPC) Recovery	Results
α Syn monomers	1:10	90%	<0.05 EU/mL
	1:100	89%	<0.50 EU/mL
	1:1000	96%	<5.00 EU/mL

Sample	Dilutions	Positive Product Control (PPC) Recovery	Results
α Syn fibrils	1:10	91%	<0.05 EU/mL
	1:100	97%	<0.50 EU/mL
	1:1000	97%	<5.00 EU/mL

Concentration of endotoxin in both α Syn monomers and α Syn fibrils was also determined using the LAL Endotoxin Assay Kit (Pierce™ Chromogenic Endotoxin Quant Kit; A39553), to confirm minimal endotoxin contamination. We confirmed that α Syn fibrils or monomers added in the culture medium (4 μ g/mL) contained endotoxin units (EU) **less than 0.5 EU/mL**, in line with the accepted concentrations (Polinski et al., 2018) to prevent triggering of an inflammatory response or induction of cell toxicity. These results have been included in the Supplementary Information.

The demonstrated lack of any significant effect α Syn monomers-treated Brain-Chips comes in support of the specificity of the experiments performed by incubating the Brain-Chip assembly with synuclein fibrils to trigger pathology (inflammation), as well as the

associated clinical relevance based on the patients' data. Aside from α Syn monomers, we have also used PBS as a negative control, as it is devoid of any associated cell toxicity.

Added text and images:

On page 10, line 229-Analysis of α Syn monomers and α Syn fibrils revealed very low endotoxin levels as measured in the culture media using the LAL assay (**Supplementary Figure 3a**).

(Supplementary Information) **On page 5, line 53-Supplementary Figure 3.** a) Detection of endotoxin levels using the Limulus amoebocyte lysate (LAL) assay. Error bars represent mean \pm SEM of n=2 measurements.

References

Polinski, N. K. et al. Best practices for generating and using alpha-synuclein pre-formed fibrils to model Parkinson's disease in rodents. *J. Parkinsons. Dis.* 8, 303–322 (2018).

Comment 4: Moreover, additional minor concerns are:

- In Fig. 1 f, the authors present the permeability to 3kDA fluorescent dextran and 0.5kDA lucifer yellow as a quantification of barrier function at different time points. However, a statistical analysis comparing the different time points is not depicted in the graph. Is the permeability at day 8 significantly lower compared to day 5?

Response: We apologize for this oversight; the revised figure (**Fig. 1f**), depicts use of a larger sample size (n=18) to strengthen our findings. As shown below, no significant changes in permeability were found between culture days 5 and 8.

Added text and images:

On page 50, line 1119- f) Quantitative barrier function analysis via apparent permeability to 3 kDa fluorescent dextran, and 0.5 kDa lucifer yellow crossing through the vascular to the neuronal channel on day 5 and 8 (n=18 independent chips, NS: Not Significant). Error bars present mean±SEM.

Reviewer #3

General Comments:

The authors have provided a substantial amount of new data and in so doing have significantly improved the manuscript. They have answered my queries and I now recommend publication.

Response: We thank the reviewer for his/her positive recommendation.

REVIEWER COMMENTS

Reviewer #2 (Remarks to the Author):

The authors have addressed most of my prior comments.

Responses to Reviewers

Reviewer #2

General Comments:

The authors have addressed most of my prior comments.

Response: We thank for the reviewer for her/his positive recommendations.